

# Modeling of discharges from Baltic Sea shipping

Jukka-Pekka Jalkanen[1], Lasse Johansson[1], Magda Wilewska-Bien[2], Lena Granhag[2], Erik Ytreberg[2], K. Martin Eriksson[2,⁑], Daniel Yngsell[2,‡], Ida-Maja Hassellöv[2], Kerstin Magnusson[3], Urmas Raudsepp[4], Ilja Maljutenko[4], Linda Styhre[5], Hulda Winnes[5] and Jana Moldanova[5]

[1]Atmospheric Composition, Finnish Meteorological Institute, Erik Palmen's Square 1, FI-00560 Helsinki, Finland
[2]Mechanics and Maritime Sciences, Chalmers University of Technology, Campus Lindholmen 41296 Gothenburg, Sweden
[3]IVL Swedish Environmental Research Institute, Lovén Center of Marine Sciences, Kristineberg, SE-451 78 Fiskebäckskil, Sweden
[4]Department of Marine Systems, Tallinn Technical University, Akadeemia Tee 15A, 12618 Tallinn, Estonia
[5]IVL Swedish Environmental Research Institute, Aschebergsgatan 44, 411 33 Göteborg, Sweden
⁑ Current address: Gothenburg Centre for Sustainable Development (GMV), Ascherbergsgatan 44, SE-41296 Gothenburg, Sweden
‡ Current address: The County Administrative Board of Västernorrland, SE-871 86 Härnösand, Sweden

*Correspondence to*: Jukka-Pekka Jalkanen (jukka-pekka.jalkanen@fmi.fi)

**Abstract.** This paper describes the new developments of the Ship Traffic Emission Assessment Model (STEAM) which enable modeling of pollutant discharges to water from ships. These include nutrients from black/grey water discharges as well as from
food waste. Further, also the modeling of contaminants in ballast, black, grey and scrubber water, bilge discharges and stern tube oil leaks are described, as well as releases of contaminants from antifouling paints. Each of the discharges are regulated by different sections of IMO MARPOL convention and emission patterns of different pollution releases vary significantly. The discharge patterns and total amounts for year 2012 in the Baltic Sea area are reported and open loop SOx scrubbing effluent was found to be the second largest pollutant stream by volume. The scrubber discharges have increased significantly
in recent years and their environmental impacts need to be investigated in detail.

## 1.   Introduction

Ship operations produce waste streams related to propulsion and engine operations, as well as crew and passenger activities (Fig 1). The waste streams related to propulsion and engine operations include bilge water from the machinery spaces, stern
tube oil from lubrication of the propeller shaft, scrubber wash water from Exhaust Gas Cleaning Systems (EGCS) for reduction of emissions of sulphur oxides into the atmosphere, ballast water from maintaining ship stability, biocides used in antifouling paints to prevent hull growth, cooling water and tank cleaning residuals. Waste streams related to humans on board include food waste, black water (sewage), and water from galleys and showers (grey water), as well as other solid waste. Operational emissions and discharges from ships are regulated through international conventions, primarily the IMO MARPOL with its



six annexes, the Ballast Water Management Convention and the Antifouling Systems Convention. These conventions regulate one "subsystem" at the time, such as bilge water production or scrubber wash water, but to assess the total impact from shipping on the marine environment it is essential to address the entire load of different stressors originating from different subsystem waste streams along with an assessment of load of species reaching the marine environment through atmospheric deposition of the shipping air pollutants. In addition to the regulation of different subsystems, some sea areas are acknowledged as *Special*

*Areas* in which, for recognized technical reasons in relation to its oceanographical and ecological condition and to their ship traffic, special rules are applied to prevent sea and air pollution. In general, the scientifically published data on the waste streams from ships is scarce, both in terms of production rates and constituents; most available data come from different types of reports (e.g. from classification societies, national authorities and intergovernmental organisations), which calls for a thorough discussion on data quality. In this paper available data on bilge water, stern tube oil, scrubber water, ballast water,

antifouling paints, food waste, black water, and grey water are collected and used to assess the input of different stressors from ships to the marine environment.

### 1.1.  Discharge inventories

In order to proceed from assessment of discharges from individual ship to assessment of the total discharge from shipping in a geographical area, it is necessary to combine the discharge factors to the activity patterns. This has previously been done for

emissions to the atmosphere and underwater noise using Automatic Identification System (AIS) with vessel technical details of the Baltic Sea fleet using the Ship Traffic Emission Assessment Model (STEAM3; Jalkanen et al., 2009, 2012, 2018; Johansson et al., 2013, 2017). The use of observed vessel traffic through AIS allows a realistic description of vessel movements and facilitates studies of vessel specific emissions which are modeled according to individual vessel features and operation. Previously, atmospheric emissions and underwater noise have been included in STEAM, but the current work extends the

model capabilities to include discharges to water.

The international conventions that regulate the waste streams into water are also important for ship emission modeling work, which also needs to follow the requirements of the conventions in order to reflect the real-world shipping as closely as possible. However, a strict application of international conventions in modeling work assumes that everyone follows the conventions, which may not always be the case (Beecken et al., 2014, 2015; Burgard and Bria, 2016; HELCOM, 2018b; Kattner et al.,

2015). In contrast to air emissions, discharges do not necessarily occur at the same locations where they are generated because their release to the sea are governed by several rules and regulations. Further, some discharges are completely random and occur whenever various holding tanks onboard are full, which makes modeling work challenging.




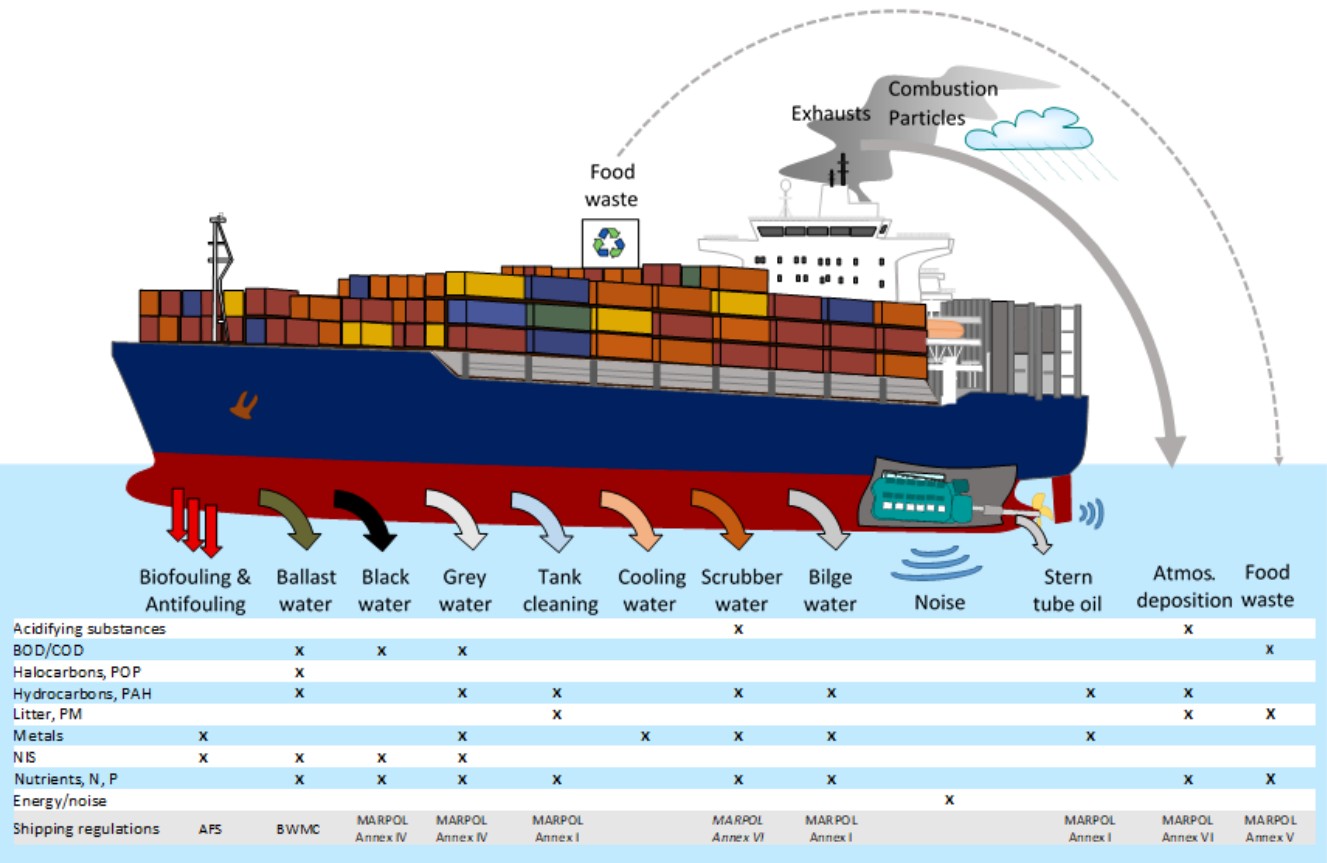

| | Biofouling & Antifouling | Ballast water | Black water | Grey water | Tank cleaning | Cooling water | Scrubber water | Bilge water | Noise | Stern tube oil | Atmos. deposition | Food waste |
|---|---|---|---|---|---|---|---|---|---|---|---|---|
| Acidifying substances | | | | | | | x | | | | x | |
| BOD/COD | | x | x | x | | | | | | | | x |
| Halocarbons, POP | | x | | | | | | | | | | |
| Hydrocarbons, PAH | | x | | x | x | | x | x | | x | x | |
| Litter, PM | | | | | x | | | | | | x | x |
| Metals | x | | | x | | x | x | x | | x | | |
| NIS | x | x | x | x | | | | | | | | |
| Nutrients, N, P | | x | x | x | x | | x | x | | | x | x |
| Energy/noise | | | | | | | | | x | | | |
| Shipping regulations | AFS | BWMC | MARPOL Annex IV | MARPOL Annex IV | MARPOL Annex I | | MARPOL Annex VI | MARPOL Annex I | | MARPOL Annex I | MARPOL Annex VI | MARPOL Annex V |

**Figure 1: Waste streams from ships and the constituents in terms of stressors on the marine environment. These releases are regulated through several international conventions, like the IMO MARPOL, Antifouling (AFS) and Ballast Water Management Conventions (BWMC). Releases of excess energy (noise, heat, light) to the sea are not currently regulated.**

Figure 1 contains an overview of various pollutant streams from ships as well as the regulatory references which are used to mitigate the environmental impacts of shipping. Various annexes of IMO MARPOL convention deal with air and water pollution, but separate conventions exist for antifouling paints and ballast water. There are also unregulated environmental pressures. These include e.g. various energy releases (noise, light, heat). Many of the existing regulations have been significantly tightened during the last decade when the shipping contribution to pollution has been quantified in detail. In the following sections, an overview of the contribution of various discharge streams is presented and relevant regulatory texts are introduced.

### 1.2. Bilge water

Bilge water accumulates in the lowest part of the ship. It has been referred to as "the mixture of water, oily fluids, lubricants and grease, cleaning fluids and other wastes that accumulate in the lowest part of a vessel from a variety of sources including





engines (and other parts of the propulsion system), piping, and other mechanical and operational sources found throughout the
machinery spaces of a vessel", (Albert and Danesi, 2011). The compounds in bilge water that are of primary concern are diesel
fuel, glycol-based coolants, and engine-, transmission-, hydraulic oils. However, trace amounts of almost everything used
onboard ships can be found in bilge water (Stamper and Montgomery, 2008). On board treatment of bilge water is focused on
the oil content, where the maximum allowed concentration to discharge according to MARPOL Annex I, is equal to 15ppm.
The release of bilge water to the sea is generally allowed everywhere if the IMO criterion is met, but some exceptions exist,
like the Antarctic and coastal areas of Finland where any oily release to the sea is forbidden.

### 1.3.  Stern tube oil

Another source of oil pollution from ships is the stern tube oil. The propeller shaft connects the main engine and the propeller
through the stern tube which goes through the ship hull. Stern tube contains bearings, sealing and lubrication system. Although
there are water-lubricated propeller shafts on the market, the most commonly used (~90% of the market, Sengottuvel et al.,
2017) lubrication is still oil-based and usually contain large number of additives (Habereder, Moore and Lang, 2009)  and seal-
improving agents like teflon and bentonite. Modern large propellers can weigh over 100 tonnes and push the propeller shaft
downwards, leading to imperfect sealing and lubricant leakage, especially if propellers or shafts experience any damage from
e.g. metal fatigue, tangled fishing nets or ice. In December 2013, Environmentally Acceptable Lubricants (EALs) became
mandatory for large ships sailing near the American coastline, but compelling regulation of this kind is not currently applied
to ships sailing the Baltic Sea. The main component of EALs is a biodegradable base, which is often vegetable oil, synthetic
ester or polyalkylene glycol (US EPA, 2011a).

### 100   1.4.  Scrubber wash water

Oil residues may also be present in scrubber wash water. Scrubbers are a type of EGCS used as an alternative to low sulphur
fuel to reduce ships' emissions of sulphur oxides ($SO_X$) to the atmosphere, according to MARPOL Annex VI. Emissions of
($SO_X$), nitrogen oxides ($NO_X$) and particulate matter (PM) into the atmosphere from shipping are substantial on local, regional
and global scale, and are associated with large societal costs in terms of affected human health as well as environmental impact
(Barregard et al., 2019; Brandt et al., 2013; Corbett et al., 2007; Endres et al., 2018; Hassellöv et al., 2013; Jonson et al., 2015;
Liu et al., 2016; Sofiev et al., 2018). Since the late 1990's cost-benefit analyses of $SO_X$ emissions from shipping, has motivated
establishment of $SO_X$ Emission Control Areas and in January 2020 new stricter global limits came to effect ((IMO, 2016)).
The regulations limit the maximum allowed fuel sulphur content (0.1% in SECA, 0.5% elsewhere). As an alternative to meet
the stricter regulations, ships can install EGCS (often referred to as wet SOx scrubbers) that cleans the exhausts with a fine
spray of seawater that is discharged back into the sea (open loop scrubber) or a spray of recirculated freshwater with added



base (closed loop scrubber). Also wet hybrid scrubber systems exists, which can alternate between open loop and closed loop mode. There are also some experimental setups of dry scrubbers, which use a granulated substrate to adsorb sulphur oxides. Wet scrubber installations are efficient in respect to the removal of $SO_X$ from the exhausts, but also to some extent wash out other substances, like $NO_X$, PM, PAH:s and metals (Turner et al., 2017; Winnes et al., 2020; Ytreberg et al., 2019)). The long

term risks for the marine environment following large scale use of wet scrubbers are today not well understood (Koski et al., 2017), yet there are an increasing number of reports arguing that scrubbers are moving an environmental impact from the atmosphere to the marine environment.

## 1.5. Ballast water

Ballast water onboard ships is used to ensure vessel stability, a proper immersion of the propeller and optimal vessel trim. Very large quantities of ballast water can be transferred between different marine regions by dry and liquid cargo ships, which take in ballast water upon cargo discharge and empty their ballast tanks when cargo is loaded to the ship. Ballast water can also be exchanged during vessel transit to reduce the risk of introducing alien species with the vessel ballast from one area to another. Since shipping is present in all sea areas non-indigenous species (NIS) are transferred between ports regardless of

differences in environmental conditions. Spreading of non-indigenous species is ranked as one of the worst threats to the marine environment by the IMO, yet it took 13 years to get the Ballast Water Management Convention (IMO, 2004) in place. The BWMC entered into force in September 2017 and requires ships to use approved ballast water treatment systems that neutralizes the organisms in the ballast water tanks, ensuring that viable harmful NIS are not transferred between ports with the ballast water. The ballast water treatment systems use either physical measures such as ultrasound or UV-light, or addition

of chemicals or to disinfect the water. The disinfection may also cause formation of unintentionally produced byproducts such as bromate.

## 1.6. Antifouling paints

Ballast water and vessel hull are two main vectors for introducing NIS by ships. Submerged structures offer substrate for

various sessile organisms such as algae, hydroids, barnacles and mussels that attach and grow on the surfaces, thereby increasing the roughness of the hull surface. Such increased roughness in turn increases drag and significantly affects the fuel consumption (Schultz, 2007) and may also affect the maneuvering capability of a ship. To reduce this fuel penalty, secure maneuvering capability and prevent spreading of NIS, the hull is coated with antifouling coatings that contain and release toxic compounds (biocides). Modern coatings use several techniques of self-polishing polymers, contact leaking systems and

controlled depletion polymers. The antifouling substance that currently dominates the market is copper oxide (Blossom, 2018). However, other substances are also used and in some coatings they are used as so-called booster biocides together with copper oxide (Yebra et al., 2004). Until 2007 it was allowed to use the toxic organometallic compound Tri-butyl-tin (TBT) on ships,

but following the adoption of the Antifouling Systems Convention, it was banned in 2008 (Champ, 2003). The first alarm reports regarding large scale negative effects from TBT came from the seafood industry as French oyster farmers lost their

harvests (Alzieu, 1991; Alzieu et al., 1986). TBT is especially harmful for crustaceans and it is accumulated in fatty tissues.

### 1.7. Food waste, black- and grey water

Discharge of food waste may cause an increased biological or chemical oxygen demand as the organic matter is degraded in the marine environment. It may also contribute to eutrophication through its nutrient content. According to IMO MARPOL

Annex V, food waste can be released to the sea if the ship is outside 12 nautical mile distance from shore, or if the waste is comminuted and passes through 25 mm mesh and is released outside 3 nautical miles from shore. In designated Special Areas, such as the Baltic Sea, all food waste must be comminuted or ground before discharge, also outside 12 nautical mile range from the shore. Similarly, untreated sewage, or black water, produced on ships is only allowed to be discharged outside 12 nautical miles from the nearest land. Treated (disinfected) black water can be discharged outside 3 nautical miles, when the

ship is en route at a speed of 4 knots or greater.

In the Baltic Sea, currently the only special designated area for sewage, all discharge of sewage from passenger ships will require on board treatment and disinfection prior to discharge. The regulations apply to new built passenger ships as from 2019 and will apply to all passenger ships as from 2021. For direct passages between St. Petersburg (Russia) and the North Sea there is an extension until 1 June 2023. Grey water consists of drainage from dishwater, showers, laundry, baths and washbasins and

is not yet regulated through the IMO. However, sometimes black and gray water waste streams are mixed in the same tank and then the regulations for black water apply.

### 1.8. Aim

The aims of this study are to **a)** expand the existing STEAM ship emission model to include a description of environmental

stressors from shipping to the marine environment. This makes it possible to **b)** construct inventories for ship discharges using the Baltic Sea as a case study area (Figure 2).

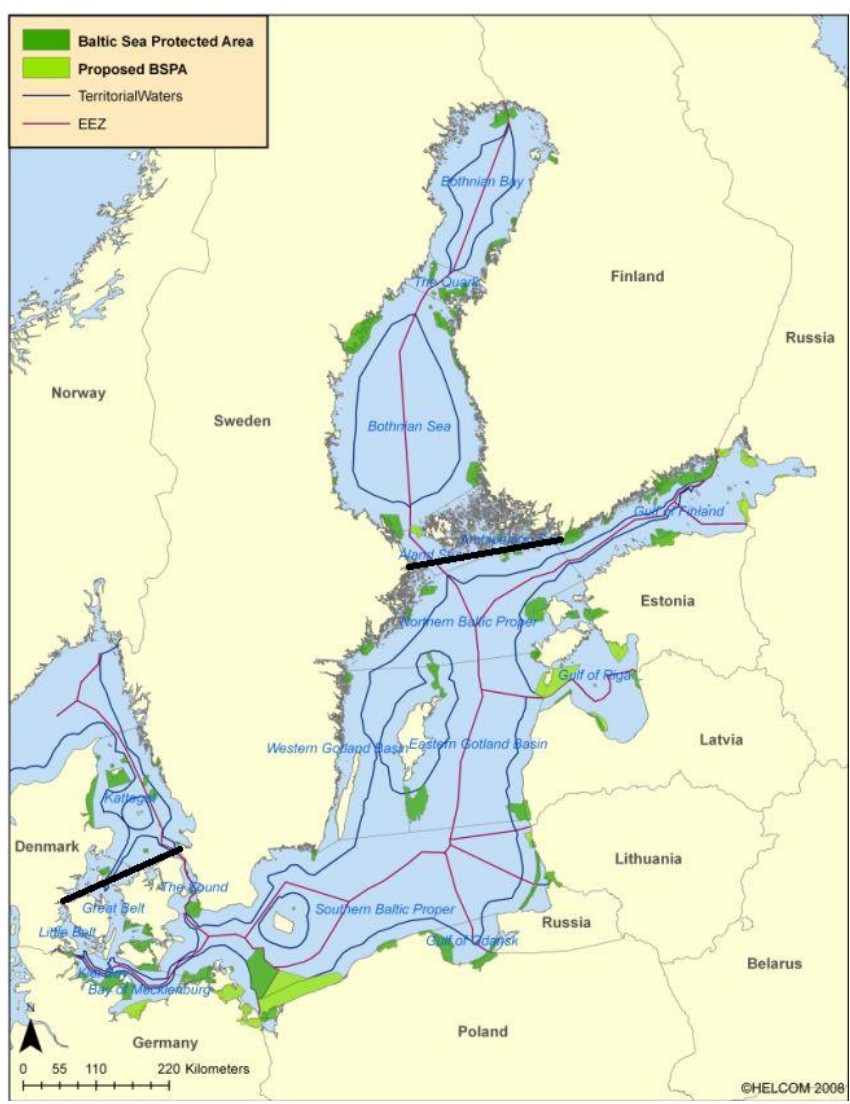

**Figure 2: The Baltic Sea and location of its sub basins. The blue line shows the border of territorial waters (12 nautical miles and the purple lines indicate the Exclusive Economic Zones. Black thick lines show the boundaries between different areas used to define antifouling classes in Table 3. Green color indicate existing (dark green) and proposed (light green) protected areas. Image from HELCOM, Baltic Sea protected areas map, 2008.**

Currently, scattered studies exist for various discharges, but data which could be used to assess the environmental impact of shipping are scarce. The methodology introduced in this paper includes description of several key pollutant streams from ships. The long-term goal of this work is to **c)** facilitate consecutive studies concerning air and water pollution, contaminants and nutrients, from ships to understand the contribution of ships in relation to other environmental stressors. Some of the recent work is already available for air and noise emissions (Jalkanen et al., 2018; Karasalo et al., 2017; Karl et al., 2019a; Raudsepp et al., 2019; Wilewska-bien et al., 2019), and regular annual reporting of discharges has been started together with the Baltic





Sea countries. These efforts aim to **d)** provide scientific background for future regulation or significant change in the existing
        conventions concerning shipping in the Baltic Sea region.

## 2.     Material and methods

        Total discharges of water pollutants from ships in the Baltic Sea area for the year 2012 were modelled using the Ship Traffic
Emission Assessment Model (STEAM3; (Jalkanen et al., 2009, 2012, 2018; Johansson et al., 2013, 2017). In the cases of
        bilge, stern tube oils, scrubber-, ballast-, black- and grey water, the discharge or leakage volume was predicted by STEAM. In
        order to obtain the mass of a specific pollutant from these waste streams, a multiplication with pollutant water concentration
        is required. These can be obtained from laboratory analyses of water samples from ships, the values used in this work are given
        in Supplementary material. Modeling the volumetric release of water from various sources facilitate the inclusion of many
pollutants when used in conjunction with pollutant concentration data from water analysis. A literature review was conducted
        to characterize the concentrations of chemicals and nutrients in the different ship waste streams. The literature review included
        both scientific reports and reports from various other sources, e.g. EPA reports and IMO documents. When a specific
        compound was reported as "not detected" in the reports, 50% of the limit of detection (LOD) or 50% the limit of quantification
        (LOQ) was used as the default value  as recommended by the US EPA (US EPA, 2000). In the case where neither LOD nor
LOQ were reported, a value of zero was used for not detected compound.

### 2.1.  STEAM v3.2

        STEAM3 software predicts instantaneous vessel power consumption based on water resistance calculations and reflects the
        variability of ship operation based on information reported in AIS. With a detailed vessel description, it is possible to produce
estimates of instantaneous power, fuel consumption and emissions. In contrast to emissions of atmospheric pollutants, the
        water discharges may not directly depend on vessel operation speed, but on other features like number of people carried
        onboard, wet surface area of ships' hull and cargo capacity. For this purpose, IHS (IHS_Global, 2016) SeaWeb vessel database
        was used to extract significant data to support discharge modeling of water pollutants. Vessel activity recorded by the AIS
        network of the Baltic sea countries was used in this work. The data consisted of position reports from year 2012 (275 million
records) which describe vessel location every 5-6 minutes. The data were obtained from the Baltic Marine Environment
        Protection Committee (HELCOM) AIS data archive. The changes listed in this paper bring STEAM to v3.2 which facilitates
        discharge modelling using the methodology described in this paper.

        It should be noted that the outputs of STEAM v3.2 describe water volumes for ballast, bilge, grey, black and scrubber wash
water and further multiplication with pollutant concentrations in these discharge waters is needed to generate inventories



which, for example, describe the copper content of scrubber wash water released to the sea. The variability of pollutant concentrations in the discharge waters may be significant, which will inevitably be reflected in the final data. Inclusion of hundreds of chemical species in various discharges was not feasible in STEAM, which necessitated the modelling of water volumes instead of reporting directly pollutant mass flux. However, we acknowledge this and produce water volume data to

facilitate further use in water research. Calculation examples and a review of existing pollutant concentration in various discharge streams are provided in Supplementary material.

### 2.2. Bilge water

Bilge water is a chemical mixture, consisting of oil residues, cleaning agents and condensed water from machinery spaces.

Estimating the volume of bilge water produced is challenging as it is not solely related to vessel activity but may depend on random occurrences of crew activities and machinery incidents, which cannot be predicted reliably. It should be noted that also stationary vessel produces bilge water in our approach, because leaks from pipes and condensation of water on metal surfaces may also occur regardless of vessel movement. The IMO MARPOL Annex I regulates when and where bilge water can be released to the sea and it is challenging to describe this in emission models. In this work, each vessel has a simulated

bilge tank and bilge water accumulated in this tank with the daily rate defined by (1a).

$$b_{pas} = 0.1313p + 373.4 \tag{1a}$$

$$b_{other} = 0.0247p + 154.4 \tag{1b}$$

The daily bilge water production [liters*day$^{-1}$] is calculated as a function of $p$, the installed main engine power. The equation for $b\_pas$ is used for passenger ships (RoPax, passenger ships and cruisers) and $b\_other$ is used for all other vessel categories.

According to DNV (Furstenberg et al., 2009), 75% of the produced bilge water is discharged overboard and the remaining share (25%) is left in port reception facilities. In the current study, this is taken into consideration by multiplying the estimated bilge water production of each ship with 0.75.

This bilge tank continuously releases bilge water to the sea in areas where this is allowed by IMO MARPOL Annex I or

national legislation (Finnish Marine Environment Protection Act 29.12.2009/1672). In reality, bilge tank is emptied more or less at random intervals, but realistic description of a random release is difficult to include, and we have chosen to implement a continuous bilge water release in this work. Based on a report by DNV (Furstenberg et al., 2009) and a study by Magnusson et al. (2018)) for 30 vessels in total, several attempts were made to correlate daily bilge water production of different kinds of vessels to build year, vessel type, size and installed main engine power. The highest score was obtained with a simple linear





model as a function of main engine power defined separately for two vessel categories since passenger vessels produce clearly more bilge water than other vessel types.

Currently, STEAM produces bilge water volumes as output and does not indicate the amount of oil or other contaminants released to the sea, because there are significant variations in pollutant concentrations of bilge water samples. In order to
generate an inventory for oily releases, water volumes need to be multiplied with measured bilge water pollutant concentrations. Bilge water can contain, besides oil residues, metals and other organic compounds. A literature search found three scientific articles (McLaughlin et al., 2014; Penny and Suominen-Yeh, 2008; Tiselius and Magnusson, 2017) and US EPA report (Albert and Danesi, 2011) where bilge water onboard ships has been sampled and analyzed for oil residues, metals and other organic compounds. Summary table of contaminants in bilge water is included in the Supplementary Material.


### 2.3.    Stern tube oil

Oil leaking from ships' stern tubes was modeled as volumes of oil released per day from different ship types (Table 1). This value was different for cargo and passenger vessels, but scant experimental evidence of actual oil release through stern tube sealing exists. Continuous oil release is assumed as a function of time, regardless of vessel activity. Although, the current
analyses, based on Etkin (2010), are more sophisticated than the leakage of over 80 million liters estimated by Canada (IMO, 2008), more experimental work is needed to determine accurate stern tube oil emission levels for ships. The approach in this study was limited because no consideration was taken to the number of stern tubes in each vessel, age of the vessel, type of sealing or the lubricant used, but instead the values from Etkin (2010) were used.

**Table 1: Leakage rates of stern tube oil for different ship types. The data was gathered from Etkin (2010).**

| Ship type | Discharge rate (liters/day) |
|---|---|
| RoPax Ship | 6 |
| Container/Ro-Ro Cargo Ship | 4 |
| Passenger Cruise Ship | 2 |
| Passenger Ferry | 2 |
| Cargo Ship | 6 |
| Refrigerated Cargo Ship | 4 |
| Container Ship | 5 |
| Chemical Tanker | 4 |
| Crude Oil Tanker | 4 |



| | |
|---|---|
| Oil Products Tanker | 3 |
| LPG Tanker | 3 |
| LNG Tanker | 1 |
| Fishing Vessel | 2 |
| Vehicle Carrier | 3 |

## 2.4. Scrubbers

In 2019, there were over 640 ships worldwide operating, and over 2000 vessels equipped, with open, closed or hybrid EGCS. With the STEAM3 model, EGCS wash water volumes for each individual ship equipped with a scrubber can be calculated.
The type of equipment installed (open, closed, hybrid system) and installation date can be used as input to the model. The total load of e.g. metals emitted is dependent on the type of scrubber system used, the engine power and wash water volume. The wash water analysis data for both open loop and closed loop systems is limited. Characterization data (concentrations of contaminants and nutrients) of scrubber wash water during open- and closed loop operations was obtained from six reports (Hufnagl et al., 2005; IMO, 2019) (MARINTEK 2006, US EPA 2011a, Kjølholt et al. 2012, PPR 6/INF.20 2018) and one
scientific article (Turner et al. 2017). These are summarized in Supplementary Material.

### 2.4.1. Scrubber operation mode

For the hybrid scrubber case, it is not known if the ship operates on open or closed mode unless the vessel travels in an area where wash water release from open loop scrubbing is not allowed (inland waters or ports in countries like Belgium, France
and Germany). However, since the water consumption will increase dramatically in low alkaline waters, we assume that the open-loop operation will be insignificant in the Gulf of Bothnia. In the Baltic Proper we assume the hybrids scrubber operating with open loop mode and switch to closed mode occurs upon entering the Bothnian Bay, where water alkalinity is low because of river runoff (Beldowski et al., 2010). Different scrubber operating modes produce very different effluent flows per power unit. Scrubber effluent releases were determined by STEAM based on instantaneous power needed to propel the vessel at
speed indicated by AIS.

Some countries, like Germany, do not allow the use of open loop scrubber in their ports. This feature has been included in the modeling by allowing the use of closed loop scrubber but use of low sulphur fuel is required from vessels equipped with open loop scrubbers when operating in German waters.





### 2.4.2. Scrubber utilization and discharge rates

In (Johansson et al 2017) it has been described how the instantaneous fuel sulphur content ($S_{Fuel}$) has been modelled based on the local regulations and vessel technical limitations. Our key assumption in the modelling of $S_{Fuel}$ is that the ship owners aim to minimize costs while complying with regulations. For the case of scrubbers, this means that the utilization level of scrubbers must depend on both the $S_{Fuel}$ used onboard and the maximum allowed $S_{Allowed}$. As an example, in an area for which 1.5% $S_{Fuel}$ is required one should not assume the same level of utilization for the scrubber than in an area in which 0.1% is required for compliance. To take this into account in the model we define scrubber utilization level, given by (Eq 2)

$$ScrubberUtil = \frac{S_{Fuel} - S_{Allowed}}{S_{Global}}; \; S_{Fuel} \leq S_{Global} \qquad (2)$$

In (2), the global average ($S_{Global}$) of 2.7 percent fuel sulphur content was assumed. For open-loop scrubbers the discharge rate has been set to equal $45m^3$ per MWh and $0.3m^3$ per MWh for closed loop respectively (Lloyds Register, 2012; Wärtsilä Corp., 2017). In the model we compute the instantaneous discharges based on the current engine power and the scrubber discharge rate, multiplied with the scrubber utilization rate. It should be noted that the use of scrubbers also affects the engine power predictions of STEAM3. The additional power required by scrubber pumps is modelled and vessel propulsion fuel consumption is increased by up to two percent (ABS, 2018), as a function of the utilization rate.

### 2.5. Ballast water

According to David & Gollasch (David, 2015), cargo ships carry ballast water amounts which correspond to 26-60 percent of their deadweight (DWT). Largest amounts can be found in bulk cargo carriers and tankers. The most reported values for the ballast water capacities range between 30-40 percent of DWT. However, modern cargo ships may visit several ports and unload only part of their cargo in any port and ballast water may be taken in or discharged during each visit. It is unlikely that all of the ballast water is loaded or unloaded during each of the port calls and this in particular makes the modelling of ballast water discharges challenging. For ballast water capacities for ships we have used ship type-dependent fractions as a function of vessel DWT. These fractions have been presented in Table 2, which are mostly in line with David & Gollasch (David, 2015) but also contain values for some other ship types.





**Table 2: Estimated volume of ballast water carried by different types of vessels, indicated with a fraction of vessel Deadweight.**

| Vessel type | Estimated Ballast Water carried, %DWT |
|---|---|
| RoRo/Passenger | 29 % |
| RoRo Cargo | 13 % |
| Vehicle carrier | 23 % |
| General Cargo | 29 % |
| Bulk Cargo | 23 % |
| Refrigerated Cargo | 33 % |
| Container vessels | 33 % |
| Chemical Tankers | 40 % |
| Crude Oil Tankers | 35 % |
| LNG Tankers | 66 % |
| LPG Tankers | 45 % |
| Oil Product Tankers | 37 % |
| Passenger ships | 61 % |
| Ferries | 67 % |
| Cruise ships | 69 % |

Ballast water management systems (BWMS) are used to kill or remove organisms within ballast water. Biocides are used during the process, but also other chemicals associated with the system either intentionally or resulting from the treatment of 315 ballast water can be discharged to the marine environment. Before the BWMS can be put out on the market they must pass an environmental risk assessment where the discharge of biocides and other chemicals to the marine environment are assessed. The risk assessment and authorization are handled by the Marine Environmental Protection Committee (MEPC), a branch of IMO. Analytical data of biocides and other chemicals present in treated ballast water from 40 different BWMS applications were reviewed based on measurement reports submitted to the IMO by member states during 2006-2015. This data can be 320 found in Supplementary material.

The modelling of ballast water discharges is challenging, and a number of safety measures are used to avoid situations where the discharge is modelled but does not occur in reality. Instead of aiming for accurate timing for ballast water intake and discharges with realistic release rates, the focus on the modelling is to obtain realistic amounts and geographical distribution for discharges. First, STEAM3.2 uses a global mapping of port areas (National Geospatial Intelligence Agency, 2014) which





define the areas where ballast water discharge is allowed to occur in the model. To further avoid the modelling of ballast water discharge that does not occur in reality, the modelled discharge is allowed to happen for vessels that are berthing, and the berthing duration must exceed two hours. Any vessel that has not been traveling in cruising mode between port visits is unable to discharge ballast water in the model. Considering these mentioned modelling rules affecting the ballast water discharges, the actual timing and rate of discharge cannot be estimated without additional information. Due to this we model an almost

instantaneous (approx. one hour) discharge for ballast water when all the above-mentioned conditions are met. This is of course, not physically realistic but will prevent a partial, incomplete discharge to occur in the modelling.

### 2.6.  Modelling of antifouling paint release

The modelling approach for biocides released from antifouling paints combined AIS data, vessel characteristics and release

rates of biocides from antifouling paints. Ship wet hull surface area is calculated with STEAM3.2 as a part of the Hollenbach resistance calculations (Hollenbach, 1998). This allows for calculation of a ship's underwater surface area (in $m^2$) based on vessel physical dimensions and hull shape. It also facilitates further development because the method applied is linked to hull wet surface area, but the model can also provide estimate of the friction term as a function of vessel speed. Once the surface area is determined for a specific vessel, release rates for the mass of substances per area and time ($g*m^{-2}*s^{-1}$) are applied.

Release rates are based on a compiled data set of 184 antifouling paint products (Gutierrez, 2015; New Zealand EPA, 2012; Ytreberg et al., 2010) and calculated in this work using the Mass-Balance Calculation method (ISO 2010). The following biocides were used: copper (inorganic species that release $Cu^{2+}$), zinc oxide (released as $Zn^{2+}$) and the co-biocides copper pyrithione, zinc pyrithione, 4,5-Dichloro-2-octyl-3-isothiazolone (DCOIT) and Zineb.

Different countries have their own regulations for antifouling paint use, varying release rates were applied for the ships

depending on which geographical area the ships were operating in according to four classes *International*, *Kattegat*, *Baltic Proper* and *Gulf of Bothnia* (Figure 2). For the class International, applied to all ships that also operate outside Baltic Sea and Kattegat, all 184 antifouling paints were used to derive average release rates of biocides. The class Kattegat is applied to all ships operating in both Kattegat and the Baltic Sea and release rates for antifouling paints approved for the Swedish west coast, Finnish, Polish and German market were used. For the class Baltic Proper, applied to ships operating in both Baltic Proper and

Gulf of Bothnia, antifouling paints approved for the Swedish Baltic Sea, Finnish, Polish and German market were included. For the class Gulf of Bothnia, applied for ships operating in Gulf of Finland only, coatings approved for the Swedish Baltic Sea and Finnish market were used.

Not all vessels use self-polishing antifouling paints, because the paint can wear off as a result of abrasion against sea ice. Vessel hulls can be kept smooth with periodic cleaning using divers and brushes. We assume that all ships operating outside

the Baltic Sea area carry antifouling paints, since there are no national regulations for international routes, and the fouling





pressure in fully marine waters is more severe compared to the brackish waters in the Baltic Sea. The same assumption is made for ships operating outside the Baltic and Kattegat region (Figure 2).

It has been estimated that the percentage of ships carrying antifouling paints in the Baltic Proper is 50% (Ambrosson, 2008) and in the Bothnian Bay it is 20% (Koivisto, 2003b). Hence, for the antifouling paint class Baltic Proper and the Gulf of Bothnia an application factor of 50% and 20%, respectively, were used. Not all paints contain all six contaminants. Hence a second application factor, which estimates the fraction of antifouling paints containing a specific biocide or booster biocide, is needed. The inorganic copper compounds and ZnO have an application factor of 100% estimating all antifouling paints to contain inorganic copper and ZnO. The booster biocides have an application factor of 0.2, assuming only 20% of all paints to contain one of the four different booster biocides. This implies that 20% of the coatings do not contain any booster biocide.

In the present work, leaching rates are applied according to Table 3 (application factors included as described above).

**Table 3: Antifouling paint leaching rates by sea area. All values given in µg/cm2/day. Note that ZnO and Cu(I)Oxide are reported as mass of Zn and Cu, which is in contrast to CuPyr, DCOIT and Zineb where the total mass or organometallic molecule is considered.**

| Region | Cu(I)oxide | Cu pyrithione | Zn oxide | Zn pyrithione | DCOIT | Zineb |
|---|---|---|---|---|---|---|
| International | 24.5 | 0.238 | 4.400 | 0.425 | 0.148 | 0.441 |
| Kattegatt | 15.507 | 0.202 | 4.633 | 0.484 | 0.154 | 0.383 |
| Baltic Proper | 7.507 | 0.101 | 2.317 | 0.242 | 0.077 | 0.192 |
| Gulf of Bothnia | 3.119 | 0.021 | 1.360 | N/A | N/A | 0.061 |

A more detailed review of daily release rates for various antifouling compounds can be found in Supplementary material.

In STEAM3, the chosen approach requires two analysis rounds over the AIS data. In the first round, areas of operation are defined for each ship. During the second calculation round, the highest leaching rate of all the vessel operational areas is applied. The release rate will be multiplied with hull wet area and time to generate a map which describes the paint residue releases. The temporal variation of antifouling release is retained, and the maps can be used as input for ecosystem modelling.

### 2.7. Food waste, black- and grey water

To assess the waste streams caused by people onboard, i.e. the amount of food waste and volumes of black and grey water and produced onboard ships, naturally depends on the number of persons onboard, which also includes the crew size. STEAM3 model uses the IHS database of vessel characteristics and the passenger capacity is one of the available information fields. For most vessels this field has not been specified. In such cases the number of passengers and crew size needs to be estimated.





### 2.7.1. Passenger and crew capacity estimation

Maximum passenger capacity values for more than 500 ships was collected using Significant Ships ((RINA, 2010)) journals. This publication series contains a comprehensive view on ship construction each year, including also the size of the crew onboard. Based on the collected data statistical models were assessed to predict the listed passenger capacities and crew size as a function of basic vessel properties. In particular, the vessel length overall (LOA) was seen to correlate strongly with the passenger capacity. The calculation methodology for passenger capacity of different types of passenger vessels is presented in

Table 4.

**Table 4: Estimation rules used for maximum passenger capacity for different vessel categories. The function shows the computation used for passenger capacity based on L (Length overall, in meters). The 'minimum' asserts the minimum value for the estimation in case the number of cabins has been specified for the vessel.**

| Type | Function | Minimum |
|------|----------|---------|
| RoPaX | $0.03L^2 + 3.7L$ | 3x cabins |
| Cruiser | $0.0113L^{2.1642}$ | 3x cabins |
| Passenger/Ferry | $10.5L$ | - |
| Yacht/Sail | $0.25*L$ | - |
| RoRo | $0.12L$ | 3x cabins |


The size of crew onboard is estimated to be $L/10 +1$, however, for RoPaX and cruisers the large number of passengers introduce additional work force on the vessel; based on the vessel data we increase the crew size by 0.2 times the passenger capacity.

### 2.7.2. Passenger count on board

For the modelling of waste discharges such as food waste and black water the maximum passenger capacity estimate needs to be converted into passenger count estimate. There are significant differences in passenger capacity utilisation between seasons and only part of the total capacity may be in use. HELCOM reports passenger capacity utilisation of 90% for cruise ships (HELCOM, 2014a). Typically, summer and end-of-the-year holidays define peak seasons of the number of passengers carried

on regular routes. This applies especially to RoPax ferries traveling in the Baltic Sea area. Average capacity utilisation was



estimated by number of passengers carried by each route for two major shipping companies which concentrate on the passenger traffic in the Baltic Sea area. The routes selected were Helsinki-Tallinn, Helsinki Stockholm, Turku-Stockholm, Stockholm-Tallinn and Stockholm-Riga. For these routes and companies operating the vessels, quarterly reported passenger counts were available for year 2014.


Vessels used in each route were identified and the number of ship crossings were determined from AIS data. This analysis revealed that, on an annual average, passenger capacity utilisation rate of regular traffic is about 50 percent for the vessels included in the analysis. During summer season and Christmas capacity utilisation is close to 100 percent, but significantly less than that outside the peak seasons. Due to the small sample size for passenger counts throughout the season, the modelling

of passenger counts is based on a static 50% estimate for the capacity utilization.

### 2.7.3. Food waste, Black and Grey Water volumes and nutrient content

The generation of grey water and black water per person and per day by different ship types was obtained from DNV (Furstenberg et al., 2009) and is presented in Table 5.


**Table 5: Volumes of Grey Water (GW) and Black Water (BW) generated by different ship types.**

| Ship types according to DNV* | Ship type | Estimated volume of GW * (L/person and day) | Estimated volume of BW * (L/person and day) |
|---|---|---|---|
| Cargo | Container | 119 | 85.9 |
| Passenger | Cruise, ferry, RoPax | 157 | 33.1 |
| Tankers | Tankers | 105 | 36.7 |
| Offshore | Supply vessels | 153 | 62 |

*Furstenberg et al, 2009

Concentrations of nitrogen and phosphorus in grey water were provided by leading providers of marine equipment (Ylimaki

J/Evac, personal communication, 2 December 2015; Wien AS/Scanship, personal communication, 2 March 2016) (Table 6). The concentration of nitrogen and phosphorus in the black water could not be used directly since it was referring to the vacuum toilets. Therefore, a fixed emission factor of 16 g N and 1.6 g P per person and day was used for black water (average estimates from the providers of marine equipment and previous work of Huhta et al., 2007). This fixed emission factor is comparable to land-based estimates of 12.5 g N and 1.4 g P per person and day found in the literature (Jönsson et al., 2005).






**Table 6: Emission factors for nutrients used in the modeling**

| Effluent | P, g/(person*day), average | N, g/(person*day), average |
|---|---|---|
| Sewage, all ships | 1.6 | 16 |
| Grey water, all ships | 1.9 | 4.4 |
| Food waste, cruise ships | 2.66 | 8.7 |
| Food waste, all other ships | 0.5 | 1.7 |
| Total nutrients, cruise ships | 6.1 | 29.1 |
| Total nutrients, other ships | 4.0 | 22.1 |

In addition to the nutrients listed in Table 6, Black and Grey Water discharges may also contain various contaminants, which are released from ships to the sea. The estimated quantities of various contaminants in Black Water were based on the onboard sampling and laboratory testing of cruise and passenger ships wastewater effluent in Alaska (ADEC, 2000, 2004, 2005, 2006, 2007, 2008, 2011a, 2011b, 2013; US EPA, 2008) and from passenger ships in the and from passenger ships in the Baltic Sea (Madjidian and Rantanen, 2011). Supplementary Material contains a summary of large number of organic and inorganic components found in Black and Grey Water discharges.

### 2.7.4. Discharge of Food waste, Black and Grey water

Today, the Baltic Sea is the only designated special area under IMO MARPOL Annex IV with respect to black water discharge. In this area, passenger ships must treat their sewage prior to discharge to the sea or the sewage must be left in port reception facilities. In the modelling of black water for this paper, which is conducted for the year 2012, the rules did not apply and hence a worst-case scenario was used where all the generated sewage onboard ships were discharged to the Baltic Sea. As many ships also mix the grey water and sewage prior to discharge, the assumption in the modelling presented in this paper is that neither grey water nor sewage are discharged if the ship is closer than 12 nautical miles from the nearest land, as required by MARPOL Annex IV. The same assumption was used for food waste (MARPOL Annex V).

In STEAM3.2, waste accumulates into a virtual tank at a constant rate which is a function of people onboard, regardless of ship operation. When a vessel travels at slow speed or it travels near the coastline, no released is allowed and the water goes to a modelled tank instead. According to IMO MARPOL Annex IV, the speed limit for discharging sewage is 4 knots. Still, the speed limit of 5 knots was chosen since this is the lowest speed in Cruise mode in our AIS data setup, which was used as a trigger for these discharges. When the IMO requirements for release are met, tank is drained continuously to the sea at a rate that is significantly larger than the generation rate (50 times the generation rate). IMO has defined (IMO, 2006) the maximum





rate for the discharges. However, the discharges are characterized as infrequent events rather than continuous streams. Due to this the produced temporal and spatial distributions for food waste, grey- and black water can be crude approximations.

## 3.  RESULTS AND DISCUSSION

To visualize the output of the STEAM3.2, the gridded datasets, maps of the annual volumes from different ship waste streams
discharged into the Baltic Sea were produced. The maps are complemented by statistics on the monthly variation of discharge volumes, the shares of the total volumes discharged from the respective ship categories and the volumes discharged to the different basins in the Baltic Sea (sections 3.2-3.7). The total mass of pollutants discharged to the water depends on the volume of discharge and the pollutant concentrations in the waste streams. Generating dedicated inventories for hundreds of contaminants is not feasible, but the approach to produce waste stream discharge volumes as output in STEAM3.2 (Table 7)
enables easier assessment of additional contaminants, as new or updated data on pollutant concentrations become available.

**Table 7: Volume discharges from STEAM for Baltic Sea shipping in 2012 (in tonnes), Cu concentration in discharge samples (micrograms per liter) and calculated total discharged mass of Cu (in tonnes) into the Baltic Sea.**

| Discharge type | Discharge volume (tons) | Cu concentration (µg/L) in each of the waste streams (95% CI) | Total mass of Cu discharged, tonnes |
|---|---|---|---|
| Ballast water | 394 000 000 | 0 | 0 |
| Bilge water | 288 000 | 68.4 (39.9 - 96.8) | 0.02 |
| Scrubber wash water (Open) | 1 520 000 | 43.0 (26.8 - 59.2) | 0.07 |
| Scrubber wash water (Closed) | 10 700 | 295 (128 - 462) | 0.003 |
| Grey Water | 5 290 000 | 0 | 0 |
| Black Water | 1 320 000 | 0 | 0 |
| Stern Tube Oil | 2 800* | 0 | 0 |

*  Converted  from  3 050 000  liters  using  density  of  0.915  kg/l  (ExxonMobil_Marine,  2003)


There are large variations in the literature data of the waste streams; both with respect to volumes and concentrations. For example a large variation can occur in generated bilge water volumes and composition, even in samples of the same vessel (Magnusson et al., 2017) and similarly large variations of e.g. metal content in scrubber wash water are reported (Turner et al.,
2017). Clearly, ballast water discharges are the largest by volume (Table 7), however, in ballast water the concentration of many contaminants is significantly lower than e.g. in EGCS effluent. There are also large differences in temporal variation of



discharges, especially concerning those which are strongly connected to passenger traffic and number of people carried onboard. This particularly concerns the black and grey water discharges as well as food waste.


Some contaminants are present in several discharge streams, for example copper (in Table 7). The different waste streams have different shares of the total copper discharged from shipping the Baltic Sea. Here, it should be noted that the spatial distribution of the discharges can vary significantly. For example, ballast water discharges are concentrated near the port areas whereas the scrubber effluent discharges occur mostly along the ship lanes. The spatial features of each of the discharges are

described in more detail in the following sections and a detailed breakdown of various discharges for different ship types is provided in the Appendix A and Supplementary material.

### 3.1. Bilge Water

According to the IMO, discharges of bilge water are allowed if the oil content of the discharge is below 15 ppm. However,

national environmental legislation in Finland prohibits the release of oily waters within 4 nautical miles from the coastline. This feature is included in the discharge modeling (Figure 3).



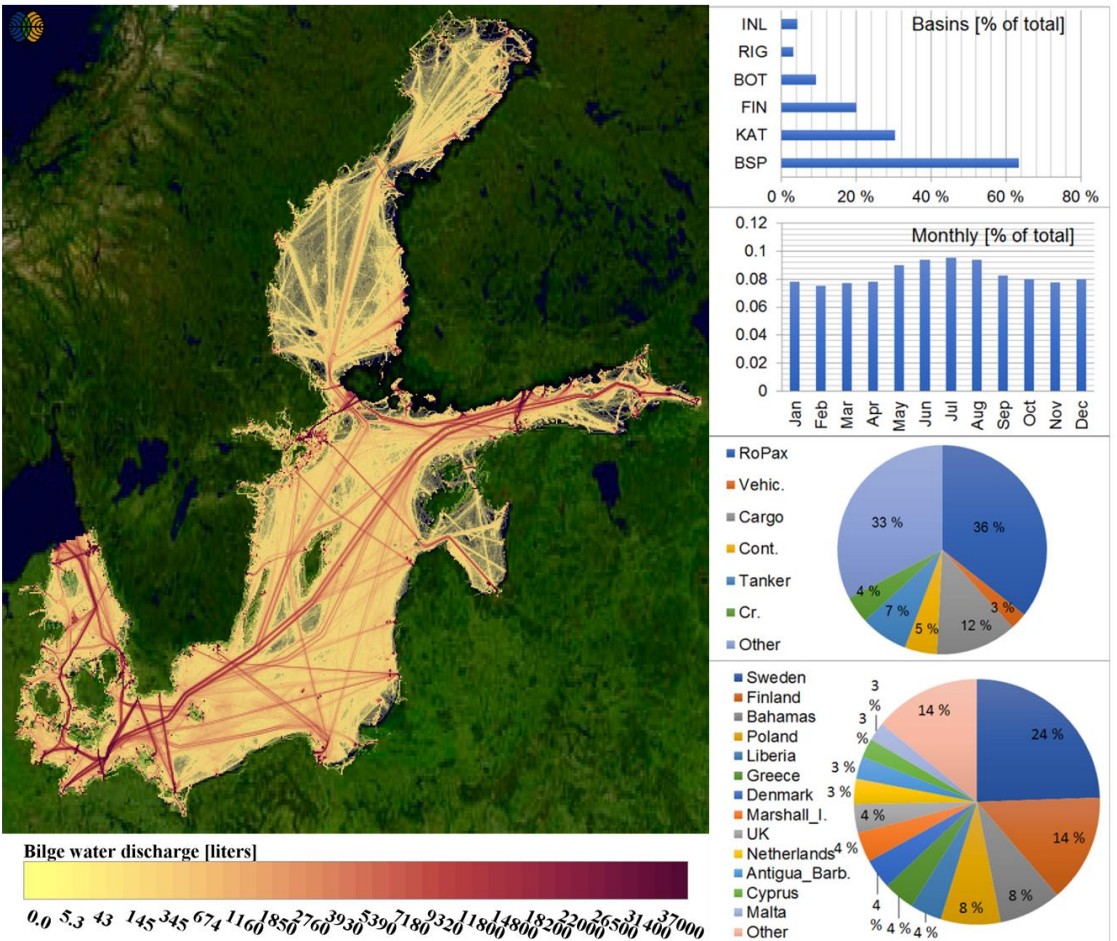

**Figure 3: Distribution of the annual bilge water releases for year 2012 from the Baltic Sea shipping. Discharge volumes are indicated**
**per map grid area of 5 km². Note, that release of bilge water is prohibited within 4 nm from the Finnish coastline. Top right: Share of bilge water released in various sub-basins of the Baltic Sea (INL=Inland areas, RIG=Gulf of Riga, BOT=Bothnian Bay and Bothnian Sea, FIN=Gulf of Finland, KAT=Kattegat, BSP=Baltic Sea Proper). Lower bar chart depicts the monthly variation of bilge water discharges. Upper pie chart summarizes the contribution of various ship types to bilge water releases and the lower pie chart the flag state shares of bilge water releases. Map background Landsat-8 image courtesy of the U.S. Geological Survey.**


The total volume of bilge water during year 2012 was 288 000 tonnes, of which 36% was from the RoPax class of vessels. According to 56 analyses of bilge water identified in literature (Albert and Danesi, 2011; McLaughlin et al., 2014; Penny and Suominen-Yeh, 2008; Tiselius and Magnusson, 2017)(see Appendix Table A1), the average effluent oil concentration was 3.2 ppm (95% CI [0.5, 6.1]), which implies that 0.9 tonnes of oil (95% CI [0.2, 2.0]) is being discharged annually to the Baltic Sea
via bilge water. Apart from oil, bilge water also contains various organic compounds and metals, and the average concentrations of 16 different PAHs, 6 other organic compounds, 16 different metals and 3 detergents are presented in the Appendix Table A1.





In reality, bilge tank is emptied more or less at random intervals, but realistic description of a random release is difficult to include, and we have chosen to implement a continuous bilge water release in this work. In principle, this will make the geographical distribution of bilge water release more diffuse than reality. It may also underestimate temporary local peaks in bilge water contaminant concentrations in the Baltic Sea. This approach does not cover the illegal discharges of untreated oily water, which were detected by aerial surveillance of the HELCOM member states (HELCOM, 2018a).

### 3.2. Stern tube oil

The annual discharge of stern tube oil to the Baltic Sea was calculated as 2800 tonnes (Table 7), which occurs during the normal operation of vessels. It should be noted that significant uncertainties may be involved. Etkin (2010) pointed out that aging may deteriorate the stern tube sealing, which would necessitate the inclusion of vessel or sealant age as a relevant modeling parameter. Nonetheless, the load of oil to the Baltic Sea is more than 3 orders of magnitude higher than what was calculated to be discharged from bilge water (0.9 tonnes). Excluding the accidental releases of oil to the sea, similar conclusion was made for the Mediterranean Sea in the EU Joint Research Centre study (Pavlakis et al., 2001). The approach taken in our study may be refined in the future, for example considering the age, number of stern tubes, type of sealing and lubricant used which are not currently considered. Figure 4 indicates the geographical distribution of predicted stern tube oil release in the Baltic Sea area during 2012.





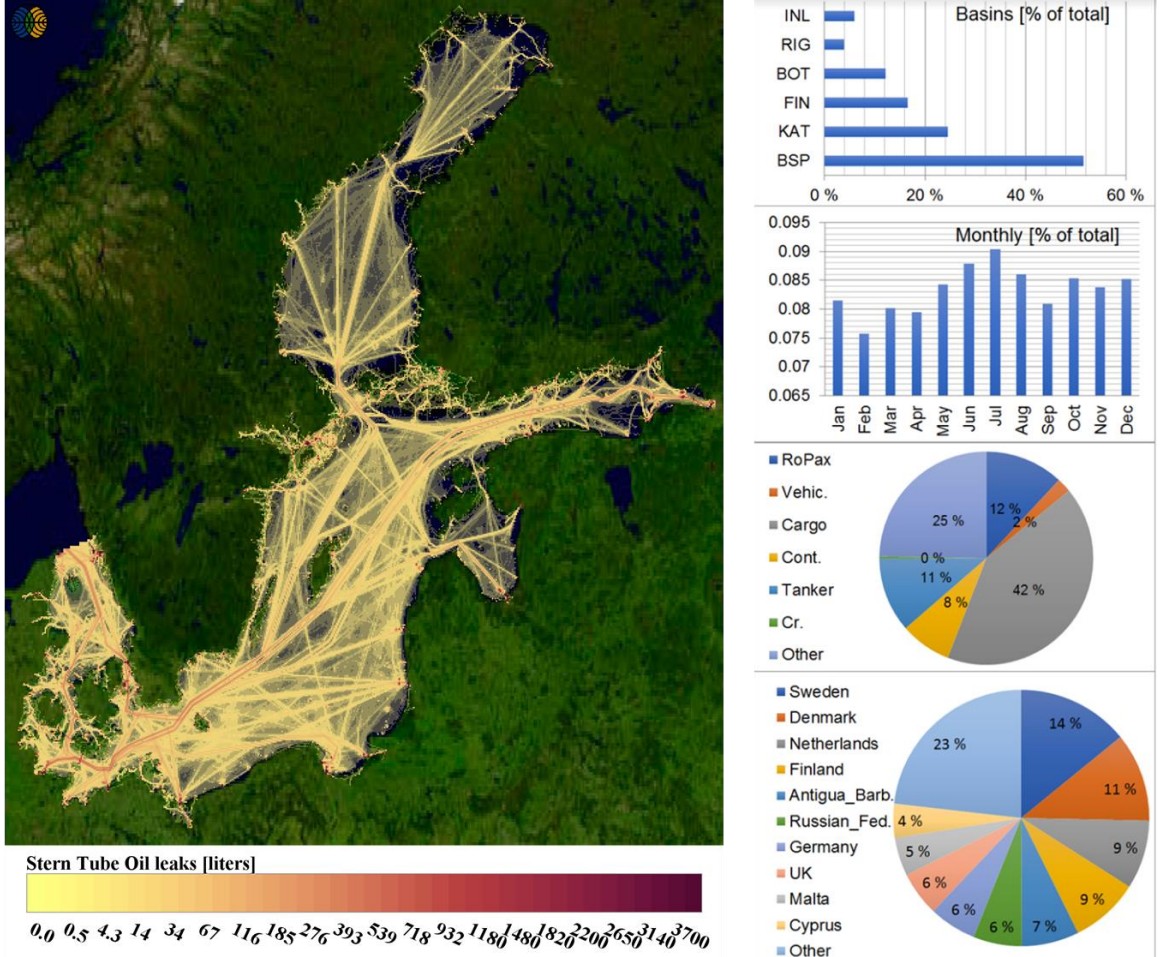

**Figure 4: Distribution of the annual stern tube oil releases for year 2012 from the Baltic Sea shipping. Discharge volumes are indicated in liters per map grid area of 5 km². Top right: Share of stern tube oil released in various sub-basins of the Baltic Sea (INL=Inland areas, RIG=Gulf of Riga, BOT=Bothnian Bay and Bothnian Sea, FIN=Gulf of Finland, KAT=Kattegat, BSP=Baltic Sea Proper). Lower bar chart depicts the monthly variation of stern tube oil discharges. Upper pie chart summarizes the contribution of various ship types to stern tube oil releases and the lower pie chart the flag state shares of stern tube oil releases. Map background Landsat-8 image courtesy of the U.S. Geological Survey.**

The geographical distribution of the stern tube oil is similar to the vessels' activity pattern. Time integration of activity will highlight areas, like ports, where ships spend significant amount of time. There exists several alternatives to stern tube oil, such as water lubrication (IMO, 2008), environmentally acceptable lubricants or airguard sealing systems to eliminate the discharge of oil to the marine environment, but as long as mineral oil lubricants are allowed to be used it will continue to be a source of oil pollution to the Baltic Sea. Most of the stern tube oil leaks occur in Baltic Sea proper and there is a maximum during the summer months. Dry cargo ships are the largest source, over three times larger than the contribution from any other ship type.





### 3.3. Scrubber effluent

Introduction of scrubbers on ships has attracted a lot of attention among the ship owners as an economic way of coping to the strict sulphur rules of the SECAs. Vessels equipped with EGCS equipment were rare until the introduction of strict 0.1% Sulphur limit in 2015. In 2012, there were only five vessels with EGCS installed; three with open loop and two with closed loop systems. In 2012, these five vessels were responsible for 1.5 million cubic meters of wash water discharge, of which >99% came from open loop systems (Figure 5 and Figure 6). A large increase of discharged water from EGCS is anticipated at a global level in 2020 when the new global sulphur cap of 0.5% is in force. Already in the Baltic Sea area, scrubber discharges have increased significantly; the 2018 estimates of scrubber discharges were estimated as 77 million cubic meters from open loop and 0.1 million cubic meters from the closed loop systems, installed in 99 vessels (Jalkanen and Johansson, 2019).

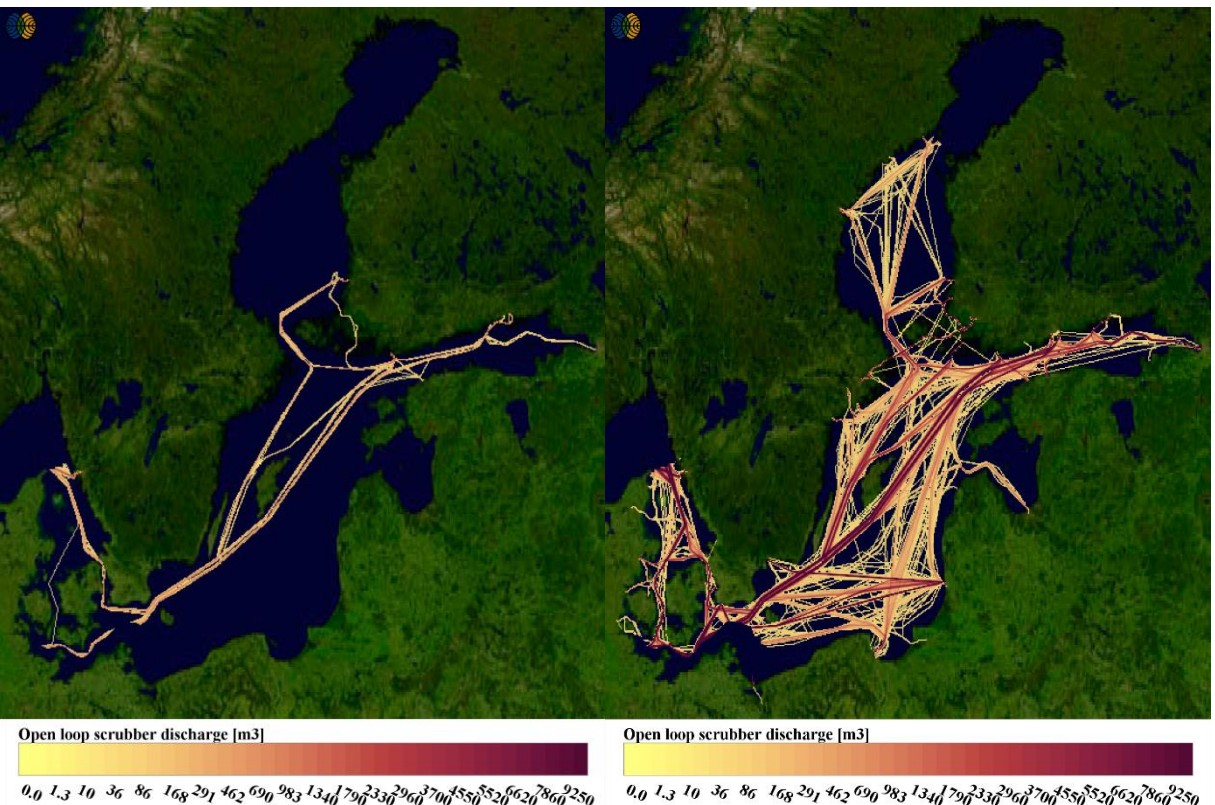

**Figure 5: Annual discharge of effluent from open loop scrubbers during 2012 (left) and 2018 (right). Both images describe the discharge in units of cubic meters per map grid cell of 5 km². A significant increase in open loop scrubber effluent release was predicted based on the activity of scrubber-enabled vessels. Map background Landsat-8 image courtesy of the U.S. Geological Survey.**



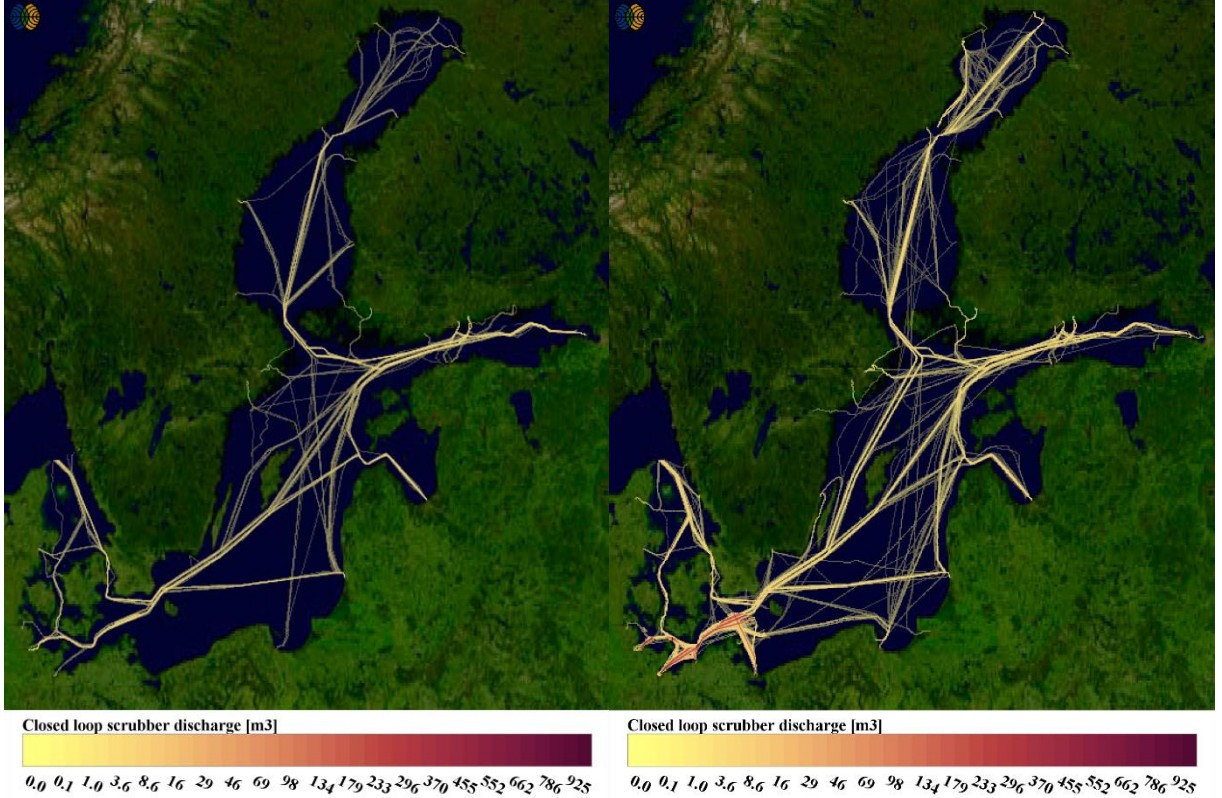


**Figure 6: Discharge of scrubber effluent from closed loop systems during 2012 (left) and 2018 (right). The values for discharges are reported in units of cubic meters for each map grid cell of 5 km². Map background Landsat-8 image courtesy of the U.S. Geological Survey.**

This is already about 25% of the ballast water discharge of 314 million cubic meters in 2018 (Jalkanen and Johansson, 2019). Discharge from open loop systems have raised concerns of water quality and have led to regional restrictions for operating open loop scrubbers. For example, German ports have prohibited the release of scrubber water and vessels should switch to low sulphur fuel instead. Similar bans have been introduced in other parts of the world (e.g China, Singapore, Malaysia), whereas some others (Japan, South Africa) have specifically allowed scrubber discharges from open loop systems.

Regional rules for the Baltic Sea open loop scrubbing are included for German waters, which are visible in Figure 5. In these areas, vessels equipped with hybrid scrubbers are operated in closed loop mode and ships with open loop systems switch to low sulphur fuels. It should also be noted that open loop scrubbing becomes more difficult in the northern part of the Baltic Sea, because of the decreasing alkalinity of brackish water. Currently, low alkalinity of the seawater is only considered in the scrubber effluent modeling for the Baltic Sea, but with proper oceanographic datasets this feature could also be implemented

globally. The literature review conducted in this paper found 56 measurements of scrubber wash water when the scrubber was operating in open-loop mode and 14 measurements in closed-loop mode (Hufnagl et al., 2005; IMO, 2018; Kjolholt et al.,





2012; Turner et al., 2017; US EPA, 2011b)(Appendix Tables A2 and A3). The data indicated higher concentration of most contaminants in wash water from closed-loop systems. For example, copper concentration was on average 295 µg/L (95% CI [128, 462] as compared to 43.0 µg/L (95% CI [26.8, 59.2] in wash water in open-loop mode (Table 7). However, assuming a

discharge rate of 45 cubic meters per megawatt-hour for open-loop wash water and 0.3 cubic meters per megawatt-hour, the load of copper per MWh is significantly higher when the ship operated in open loop mode (1.9 g/MWh) as compared to closed-loop mode (0.06 g/MWh). Figure 7 illustrates the development of scrubber effluent discharge from ships during the time period 2006-2018 (Jalkanen and Johansson, 2019). It should be noted that the scrubber discharge volumes have been increased by almost two orders of magnitude, which makes them currently the second largest volumetric discharge from ships.


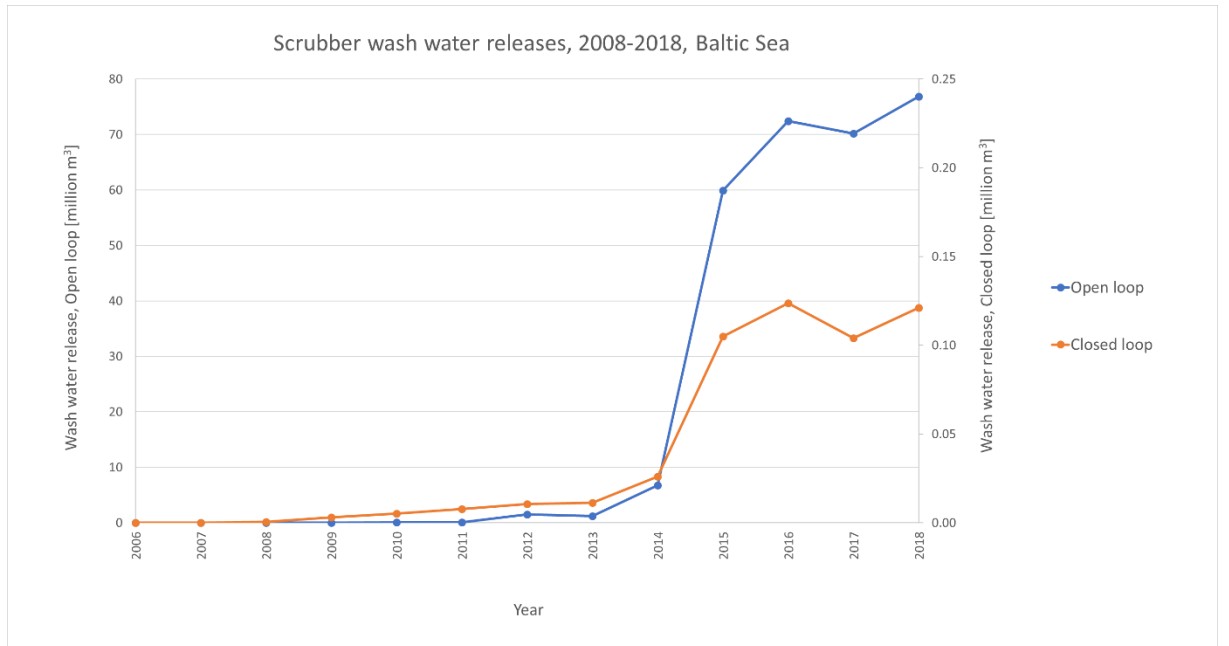

**Figure 7: Scrubber effluent discharge from the Baltic Sea fleet during 2006-2018. Image taken from HELCOM discharge report (Maritime19/13-4.INF).**

In December 2019, over 2000 vessels were included in the list of scrubber installations (IHS_Global, 2016). If a significant part of this group of vessels operates in enclosed areas, it may create problems for some marine species (Magnusson et al., 2018b). The rapid adoption of scrubbers with the continued use of high sulphur fuel were predicted as a compliance option to the IMO decision on global ship sulphur fuel reduction starting in Jan 1[st] 2020 (Faber et al., 2016).

### *3.4. Ballast Water*

The total annual load of ballast water to the Baltic Sea was modelled as 394 million tonnes in 2012. Tankers discharge the largest volume of ballast water (146 million tons) followed by dry Cargo ships (103 million tons). Discharge pattern of ballast





water is very different from other discharges (Figure 8), because ballast water operations occur mostly during cargo operations,
which indicates significant discharge volumes near port areas. Water is taken in as ballast during cargo discharge to maintain

vessel stability and proper immersion of the propeller, to be discharged when cargo is loaded to the vessel.

**Figure 8: Distribution of the annual ballast water releases for year 2012 from the Baltic Sea shipping. Discharge volumes are indicated in tonnes per map grid area of 40 km². Note the change of map resolution to highlight ballast water discharge locations on this map. Top right: Share of ballast water released in various sub-basins of the Baltic Sea (INL=Inland areas, RIG=Gulf of Riga, BOT=Bothnian Bay and Bothnian Sea, FIN=Gulf of Finland, KAT=Kattegat, BSP=Baltic Sea Proper). Lower bar chart depicts the monthly variation of ballast water discharges. Upper pie chart summarizes the contribution of various ship types to ballast water releases and the lower pie chart the flag state shares of ballast water releases. Map background Landsat-8 image courtesy of the U.S. Geological Survey.**




In this work, a description of ballast water discharge volume and pattern were identified, to facilitate future work on alien invasive species. However, this would require identification of the location where ships have filled their ballast tanks, which was not included in this work. Regardless, our results can be used to assess the risk of alien species transfer if suitable water analysis is available. Previous estimates of 250 million tonnes of ballast water discharge have been reported earlier, based on
earlier traffic estimates during year 2011 (HELCOM, 2014b), whereas our modeling estimate is 1.6 times that volume. There can be several contributing factors to this discrepancy. Currently, our discharge estimate has not established the link to cargo flow recorded at ports by customs authorities but is rather based on vessel capacity instead. Further, the modeling assumes a complete discharge of all ballast water during port visits, which will lead to an estimate which is larger than in reality. Vessels may discharge part of their ballast water if vessel takes in cargo which is less than the full cargo capacity. Our current estimates
of ballast water discharge may thus lead to a slightly higher risk of introducing alien species to the Baltic Sea area than in reality.

To minimize the risk of spreading alien species via ballast water emissions, IMO adopted in 2004 the Ballast Water Management Convention (BWMC), which entered into force in September 2017 and requires ships to manage their ballast water in such manner that aquatic organisms are removed or killed prior discharge. The installation of the Ballast Water
Management Systems (BWMS) depends on when the ship is scheduled for IOPPC (International Oil Pollution Prevention Certificate) renewal, but all ships must at the latest have an approved BWMS installed by 2024. The BWMS usually relies on filtration followed by UV radiation, biocidal treatment, deoxygenation and electrochlorination (King et al., 2012). Given that 394 million tonnes of ballast water is discharged into the Baltic Sea annually (in 2012), the requirement to treat ballast water is a new waste stream of contaminants to the Baltic Sea. In this work, the concentration of contaminants in 40 different BWMS
were compiled and is presented in Supplementary information with references to original data.

The spatial distribution of ballast water discharge indicates that these operations occur in ports. In reality, ballast water can be exchanged also during transit and should be done in quantities which are three times the required volumes to ensure proper tank flushing (IMO, 2004). However, the BWMC requires that water exchange is to be done in locations where the distance to the nearest land is at least 200 nautical miles and the depth of the sea is more than 200 meters. These requirements are
impossible to meet in the Baltic Sea area and current HELCOM recommendation is to conduct ballast operations in ports.

### 3.5. Antifouling Paint releases

Over 280 tons of Cu is being released to the Baltic Sea from ships coated with antifouling paints annually (Table 8). According to HELCOM, the waterborne input of copper is 890 tons annually and comprises both natural and anthropogenic sources
(antifouling paints excluded) (HELCOM, 2011). Monitoring data in Swedish coastal waters have shown 20 out of 36 assessed water bodies to have copper concentration that exceed the Swedish water quality criteria used for Cu in the Baltic Sea (SWAM, 2018). Hence, the load of copper from ships coated with antifouling paint is significant.





*Table 8: Emission totals for antifouling paints in the Baltic Sea area during 2012. These numbers contain the contribution from the commercial fleet.*

| Compound | Release(kilograms) | Molecular mass (g/mol) | CAS number |
|---|---|---|---|
| Cu (inorganic) * | 281 000 | 53.546* | 1317-39-1 (CuO) |
| Cu pyrithione | 575 | 315.86 | 154592-20-8 |
| Zn (inorganic) * | 55 600 | 65.38* | 1314-13-2 (ZnO) |
| Zinc pyrithione | 1 090 | 317.7 | 13463-41-7 |
| DCOIT | 371 | 282.2 | 64359-81-5 |
| ZINEB | 1 070 | 275.8 | 12122-67-7 |

* Inorganic CuO and ZnO are reported as mass of Cu and Zn, not their oxides. For CuPyr, DCOIT and Zineb molecular masses are used in mass reporting.

Another large source of copper to the Baltic Sea is antifouling contribution from leisure boats, which are not reported in this work, but were recently reported elsewhere (Johansson et al., 2020). The wet surface area of the Baltic Sea commercial ship fleet is about 44 million square meters while the corresponding area for boats is about 7 million square meters. Although, the wet surface area of a small boat is significantly smaller than that of a ship, there exist about sixty times more boats than ships in the Baltic Sea area. Small boats are primarily used during summer and are taken out of the water and stored for winter months, whereas the ship fleet operates throughout the year. Further, the activity patterns of small boats and commercially operated ships are different; small boats tend to move close to the shoreline whereas big ships follow the shipping lanes (Figure 9).





**Figure 9: Geographical distribution of the emissions of Cu(I)O from ship hull antifouling paints. The values reported by the color scale indicate CuO releases in unit of grams per map grid cell of 5 km². Top right bar chart indicates the share of Cu(I)O releases by sea area (INL=Inland areas, RIG=Gulf of Riga, BOT=Bothnian Sea and Bothnian Bay, FIN=Gulf of Finland, KAT=Kattegat, BSP=Baltic Proper). Lower bar chart describes the monthly variation of CuO releases. Upper pie chart illustrates the estimated share various ship types and the lower pie chart the flag state contributions to CuO releases. Map background Landsat-8 image courtesy of the U.S. Geological Survey.**

The spatial copper (CuO) release distribution from antifouling paints characterizes general shipping activity across the Baltic Sea. The emission increases gradually toward the southern Baltic Proper as shipping traffic from different ports merge into main shipping lines. The annual leaching of AFP along at the concentrated shipping lines ranges from 3.2 kg km⁻² in Northern





Quark strait (separates Bothnian Bay from Bothnian Sea, see Figure 2) up to 70 kg km$^{-2}$ in the Øresund Strait.  There are
distinctive hotspots in and near the port areas where annual AFP releases can be an order of magnitude higher than at shipping
lines.

The AFP emission totals reported in this paper reflect the values obtained for the commercial fleet, only, and it has been
assumed that the leaching rate of the hull paint remains constant regardless of the movement of the vessel. Strictly speaking,
this may underestimate the paint release rate, especially in cases where recently painted hull surface is exposed to water
(Kojima et al., 2016). However, the leaching rates for surfaces which have been exposed to sea water stabilize over time and
leaching rates of surfaces with paint layers older than one month still show some dependency on speed, but the differences are
much smaller than in cases of fresh paints. Recent studies conducted in the Baltic Sea and the more saline Swedish West coast
have also observed that leaching rates of copper increase with increasing salinity (Lagerström et al., 2018, 2020; Ytreberg et
al., 2017). Temperature is another parameter known to control the leaching of copper as well as the presence of biofilm on the
coating (Valkirs et al., 2003). The approach taken in this manuscript (and STEAM model) allows future work with primarily
speed, salinity and temperature dependent antifouling paint releases. Maps for other antifouling paint residues can be found in
Supplementary material.

### 3.6. Food Waste, Black Water and Grey Water


The spatial distribution (Figure 10) of food waste discharge is concentrated on areas outside the 12 nautical mile distance from
shore. Over 90% of the nitrogen in food waste comes from passenger ships, which carry many people onboard. Most of the
food waste release happens during the summer, when the cruise vessel traffic activity is high. The summer period is also a
maximum of the air emissions from ships, and part of the nutrients enter the Baltic Sea through air. Air emissions from Baltic
Sea ships have been discussed in Karl et al. (Karl et al., 2019a, 2019b) and are therefore not discussed here.





**Figure 10: Food waste nitrogen discharges from ships during 2012 in unit of grams per map grid cell area of 5 km². According to IMO rules, no discharge is allowed closer than 12 nm distance from the coastline. Upper bar chart describes the contributions by**
**sub-basin, lower bar chart the monthly variation. Upper pie chart depicts the ship type and lower pie chart the flag state contributions to foodwaste nitrogen releases. Map background Landsat-8 image courtesy of the U.S. Geological Survey.**

In reality, many vessels pump their wastewater to port reception facilities, but there also exists a few of those which pumps
everything into the sea. Our modelling approach is likely to result in overestimation of nutrient release to some regions of the sea because of this feature. However, we have reduced the overall release rate according to the fraction of ships which leave





their waste in ports during harbor stays. This will spread the sewage releases throughout the whole area where discharge is allowed, but it may lead to a more realistic description of discharge totals than the assumption where everything is released into the sea and nothing is left in port reception facilities (Figure 11). In reality, there exists vessel routes where all waste is

left in port, but our current knowledge of vessel operations does not allow the inclusion of this feature. From 2021 onwards, all passenger vessels must leave their sewage in port reception facilities (HELCOM, 2014a), whereas for cargo ships waste release is still allowed according to the IMO regulations.





**705** **Figure 11: Sewage nitrogen discharges from ships during 2012 in unit of kilograms per map grid cell area of 5 km². According to IMO rules, no discharge is allowed closer than 12 nm distance from the coastline. Upper bar chart describes the contributions by sub-basin, lower bar chart the monthly variation. Upper pie chart depicts the ship type and lower pie chart the flag state contributions to sewage nitrogen releases. Map background Landsat-8 image courtesy of the U.S. Geological Survey.**

**710** The estimated total reduced nitrogen (M=14 g mol$^{-1}$) input to the sea from black water and food waste were 556 and 82 tonnes in 2012, respectively. From these totals, 487 and 76 tonnes were from passenger vessels. In the case of phosphorus, food waste, sewage and grey water contributions were estimated as 24, 42 and 40 tonnes for the whole Baltic Sea fleet as reported earlier (Wilewska-bien et al., 2019).





From 2021 onwards, most of the passenger ship fleet operating in the Baltic Sea area will not be able to release their sewage
to the sea, but there exists an extension until year 2023 for vessels operating between the North Sea and St. Petersburg, Russia.
All sewage must be discharged to the port reception facilities from that date on, but this requirement only applies to passenger
vessels, but passenger vessels are responsible for 88% of the nitrogen in sewage and over 90% of the nitrogen in food waste
releases to the sea. This decision alone will reduce the nitrogen release from 556 to 69 tons (reduced nitrogen) and increases
the wastewater reception of ports by one million tonnes. According to Marpol Annex IV, Similar requirement does not apply
to food waste release from ships, which would further reduce the annual nitrogen load by 76 tons. Currently, there is no viable
enforcement practice in place to control whether these rules are followed.

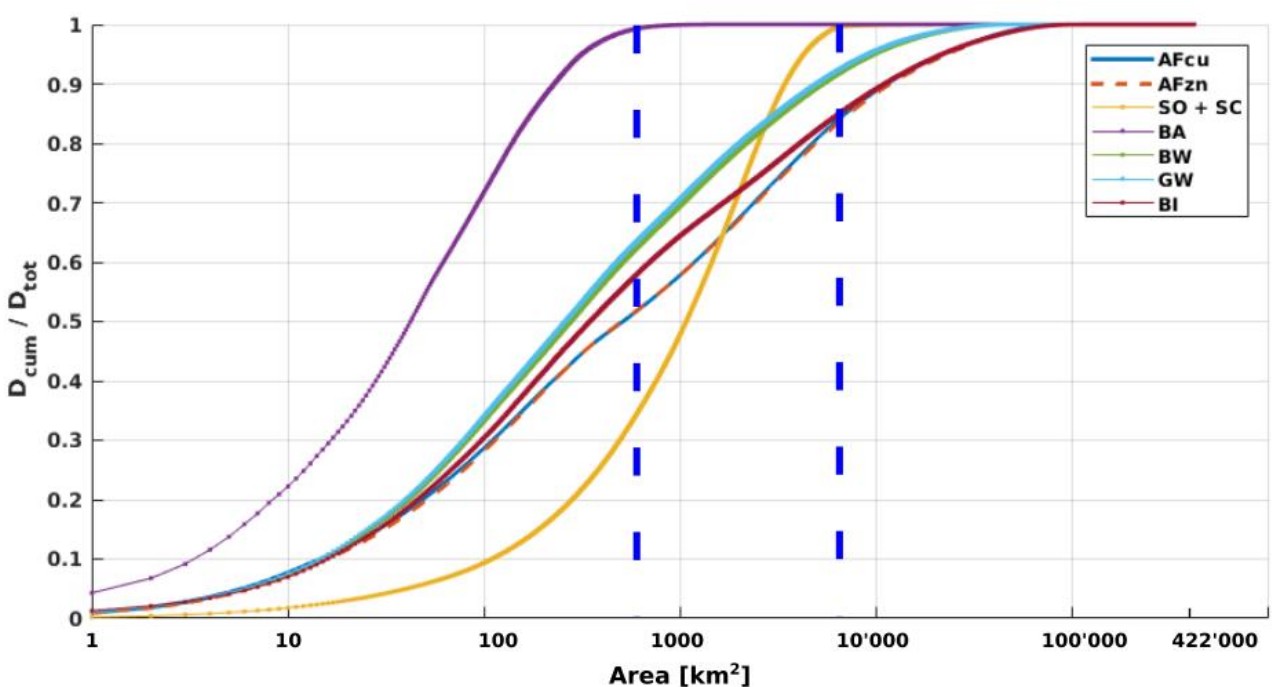

**Figure 12: The cumulative discharges over the area of the Baltic Sea divided by the total discharges for the year 2012. Vertical blue**
**dashed lines correspond to the areas of total load of the ballast water and scrubber water discharges.**

The normalized cumulative distribution of annual discharges from different waste streams as the function of Baltic Sea surface
area summarizes spatial distribution of the sources (Figure 12). Ballast water is discharged as point sources with the highest
load at a single location accounting for about 4% of the total load. Total area of ballast water discharges is less than 500 km$^2$
which accounts less than 0.12% of the total area of the Baltic Sea. Scrubber wash water is discharged along main shipping
lanes without significant hotspots. This is represented by slow increase of the cumulative distribution curve at small area
values. Rapid increase of the cumulative load between 500 km$^2$ and 6300 km$^2$ accounts for the major shipping lane discharges.





Antifouling paint contaminants, bilge, grey water and sewage discharges consist of point sources, ship lanes and low traffic area distributions covering a large part of the Baltic Sea. In the case of mixed loading, approximately 50-60% of antifouling, bilge, black water and greywater discharge can be related to hotspots, ~25% to the major shipping lines and the remaining 10-15 % to the low traffic areas. Greywater and black water have similar spatial discharge pattern represented by overlapping curves. Similar spatial distribution holds for Cu and Zn antifoulants as all ships leach both contaminants simultaneously. Black water and greywater point sources represent large discharges when the ships have passed 12 nm restriction zone.

## 4. Summary

In this paper, we have reported the methodology which was used to add a new capability to the existing ship emission model STEAM; discharges of waste streams and release of antifouling paints to the marine environment. These enable new studies on discharges to the sea from ships, based on the realistic vessel activity and technical description. In contrast to the atmospheric exhaust emissions, water discharges are released directly to the sea and contain various contaminants and nutrients. Also, atmospheric emissions occur at places where fuel combustion occurs, which is different from discharges. There exist large group of regulations which define which releases are allowed in various areas. This complicates the modeling work. Modeling of water pollutants from ships requires an analysis of maritime environmental law in various countries, because several exceptions exist to the IMO requirements. The work reported here covers most of the annexes of IMO MARPOL convention, includes antifouling and ballast water management convention rules and enable us to extend the evaluation of environmental impact of shipping beyond atmospheric studies. Some of these exemptions concern specific areas or countries (bilge water releases), are address specific technologies (open loop scrubbers) or other regional rules (sewage release restrictions for passenger vessels). We have attempted to include these requirements in this work as much as possible, but the current approach assumes that all ship comply with every rule which may not always be the case. For example, compliance rate to the sulphur rules is high, but not necessarily one hundred percent.

Strong increase of scrubber effluent releases was to be expected after the sulphur rule change in SECAs from 2015 onwards. This will probably happen also in global scale after the global 0.5% sulphur regulation starts. Significant increase of scrubber effluent release underlines the need to conduct a thorough impact assessment of this new pollution source to marine environment. For this purpose, ecotoxicological studies are urgently needed to get a holistic view and assess whether changes to sulphur reduction methods are necessary. Antifouling paints were identified as a major source of copper from ships. Since the ban of TBT in hull paints, other organometallic compounds were developed to replace tin, but further work is needed to learn how big a problem high copper releases are in marine environment.

This work fills some of the gaps in knowledge of quantities of ship generated water pollutants as we have taken first steps towards this direction. This paper lists discharges from the Baltic Sea ship fleet during the calendar year 2012. It also identifies several research topics which need further attention in the future in order to reduce the uncertainties involved in the discharge



modeling. Some of these contributions will be reported in the following papers, like the small boat emission modeling, water and air dispersion studies, but others require further experimental work to reduce the uncertainties involved in the modeling work.

**Appendices:** Summary tables for contaminants in various discharge streams are given in Appendices. Calculation examples
are also provided for antifouling paint releases, bilge and food waste nutrient discharges.

**Data availability**

The data described in this paper is available for further research. These consist of gridded binary data describing all generated daily discharge quantities. These datasets are available from Zenodo, DOI:10.5281/zenodo.4063643.


**Supplementary material**

Additional maps, a review of water analysis results for various discharge streams and spreadsheet summary of discharge totals by sea area are presented as supplements to the manuscript.

**Author contributions:**

J-PJ was in charge of the manuscript writing and contributed to STEAM discharge modelling algorithm development as well as data analysis. LJ was responsible for STEAM model development and programming. LG contributed to ballast water modeling definitions. EY was responsible for antifouling modelling work and data analysis. KME, I-MH, DY, KM and MWB contributed to the bilge, black and grey water data analysis. KM, LS, HW, JM provided expertise and data for the scrubber
washwater modelling. UR and IM were responsible interfacing STEAM with a water dispersion model, analysis of geographical distribution of pollutants and background data analysis. All authors have contributed to manuscript writing.

**Acknowledgements**

This work started in the BONUS SHEBA project and it was supported by BONUS (Art 185), funded jointly by the EU, the
Academy of Finland, Estonian Research Council, Swedish Agency for Marine and Water Management, Swedish Environmental Protection Agency and FORMAS. We are grateful to the HELCOM member states for allowing the use of HELCOM AIS data in this research. The support from EU H2020 project EMERGE is appreciated, which has has received funding from the European Union's Horizon2020 research and innovation programme under grant agreement #874990. This work reflects only the authors' view and INEA is not responsible for any use that may be made of the information it contains.



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




**Appendix A**: To obtain the mass of contaminants and nutrients, the water volumes from STEAM3.2 need to be multiplied with the results of water analysis (mass/volume). The concentration values used in this work are given in the tables below.

**Table A1. Concentration of contaminants and nutrients in bilge water. The complete dataset and references are presented in Supplementary material.**

| Contaminant | Number of ships in the analyses | Total number of analyzed bilge samples | Average concentration (µg/L) | Median concentration (µg/L) |
|---|---|---|---|---|
| Oil index Total | 18 | 40 | 6700 | 1720 |
| Oil index Fraction >C10-C12 | 12 | 17 | 177 | 61.5 |
| Oil index Fraction >C12-C16 | 12 | 17 | 578 | 184 |
| Oil index Fraction >C16-C35 | 12 | 17 | 6110 | 751 |
| Oil index Fraction >C35-<C40 | 12 | 17 | 1370 | 220 |
| naphthalene | 12 | 17 | 49.6 | 7.73 |
| acenaphthylene | 12 | 17 | 0.377 | 0.166 |
| acenaphthene | 12 | 17 | 1.470 | 0.385 |
| fluorene | 12 | 17 | 3.28 | 0.825 |
| phenanthrene | 12 | 17 | 3.460 | 0.915 |
| anthracene | 12 | 17 | 0.326 | 0.150 |
| fluoranthene | 12 | 17 | 0.769 | 0.089 |
| pyrene | 12 | 17 | 1.46 | 0.315 |
| benz(a)anthracene | 12 | 17 | 0.220 | 0.019 |
| chrysene | 12 | 17 | 0.227 | 0.041 |
| benz(b)fluoranthene | 12 | 17 | 0.130 | 0.012 |
| benz(k)fluoranthene | 12 | 17 | 0.055 | 0.010 |
| benz(a)pyrene | 12 | 17 | 0.147 | 0.015 |
| dibenzo(ah)anthracene | 12 | 17 | 0.050 | 0.010 |
| benzo(ghi)perylene | 12 | 17 | 0.175 | 0.013 |
| indeno(123cd)pyrene | 12 | 17 | 0.081 | 0.010 |
| PAH, sum 16 | 12 | 17 | 60.7 | 11.7 |



| | | | | |
|---|---|---|---|---|
| PAH, sum carcinogenic | 12 | 17 | 0.613 | 0.109 |
| PAH, sum other | 12 | 17 | 60.7 | 11.7 |
| PAH, sum L | 12 | 17 | 50.6 | 8.70 |
| PAH, sum M | 12 | 17 | 9.02 | 3.50 |
| PAH, sum H | 12 | 17 | 0.718 | 0.119 |
| Anionic surfactants | 15 | 36 | 5140 | 2330 |
| Cationic surfactants | 7 | 7 | 909 | 440 |
| Nonionic surfactants | 7 | 7 | 2530 | 400 |
| benzene | 9 | 12 | 1.91 | 0.680 |
| toluene | 9 | 12 | 11.2 | 2.54 |
| ethylbenzene | 9 | 12 | 4.59 | 2.38 |
| m,p-xylene | 9 | 12 | 21.4 | 14.0 |
| o-xylene | 9 | 12 | 193 | 8.54 |
| xylenes, sum | 9 | 12 | 388 | 31.0 |
| Aluminum | 10 | 27 | 274 | 73.5 |
| Antimony | 4 | 21 | 5.17 | 3.57 |
| Arsenic | 10 | 27 | 55.2 | 10.4 |
| Barium | 8 | 25 | 95.8 | 44.7 |
| Cadmium | 4 | 21 | 3.75 | 1.00 |
| Calcium | 10 | 27 | 89300 | 79200 |
| Chromium | 9 | 26 | 20.8 | 3.88 |
| Cobalt | 4 | 21 | 3.88 | 2.26 |
| Copper | 10 | 27 | 126 | 115 |
| Iron | 6 | 23 | 2610 | 1071.5 |
| Lead | 9 | 26 | 8.62 | 7.5 |
| Magnesium | 10 | 27 | 214000 | 183000 |
| Manganese | 10 | 27 | 100 | 56.1 |
| Nickel | 9 | 26 | 31.2 | 17.0 |
| Potassium | 8 | 25 | 82600 | 68900 |
| Selenium | 7 | 24 | 25.2 | 25.1 |
| Sodium | 8 | 25 | 1800000 | 1450000 |
| Vanadium | 5 | 22 | 53.3 | 38.4 |
| Zinc | 10 | 27 | 568 | 151 |





| | | | |
|---|---|---|---|
| Total phosphorus | 7 | 13 | 5.4 | 4.2 |
| Total nitrogen | 4 | 10 | 12.4 | 14.2 |
| Ammonia | 3 | 3 | 2.63 | 1.63 |
| Total Kjeldahl nitrogen | 3 | 3 | 0.866 | 0.71 |



**Table A2. Concentration of contaminants and nutrients in open loop scrubber wash water. Note, that full references to**
**sample analysis reports are given in the Supplementary material.**

|  | #Samples included | Average concentration (µg/L) | 95% CI lower (µg/L) | 95% CI upper (µg/L) |
|---|---|---|---|---|
| Arsenic | 41 | 7.9 | 2.6 | 13.2 |
| Barium | 3 | 68.3 | -150.4 | 287.1 |
| Cadmium | 38 | 1.0 | 0.5 | 1.5 |
| Calcium | 8 | 394875 | 383980 | 405770 |
| Chromium | 34 | 14.5 | 9.0 | 20.0 |
| Copper | 47 | 43.0 | 26.8 | 59.2 |
| Iron | 1 | 93.0 |  |  |
| Litium | 8 | 44385 | -24116 | 112886 |
| Lead | 44 | 11.8 | 5.1 | 18.6 |
| Manganese | 1 | 20.0 |  |  |
| Magnesium | 8 | 1223875 | 1186035 | 1261715 |
| Mercury | 19 | 0.1 | 0.1 | 0.1 |
| Nickel | 42 | 51.8 | 36.6 | 67.0 |
| Potassium | 8 | 377750 | 355511 | 399989 |
| Selenium | 2 | 97.0 | 58.9 | 135.1 |
| Strontium | 8 | 3012743 | -456605 | 6482092 |
| Vanadium | 42 | 179.2 | 123.0 | 235.4 |
| Zinc | 45 | 119.1 | 25.7 | 212.4 |
| Naphthalene | 35 | 3.6 | 2.5 | 4.7 |
| Acenaphthylene | 34 | 0.1 | 0.1 | 0.2 |
| Acenaphthene | 34 | 0.3 | 0.2 | 0.4 |
| Fluorene | 34 | 0.6 | 0.5 | 0.8 |
| Phenanthrene | 35 | 1.8 | 1.3 | 2.3 |
| Anthracene | 34 | 0.1 | 0.0 | 0.2 |
| Flouranthene | 34 | 0.2 | 0.1 | 0.3 |
| Pyrene | 34 | 0.4 | 0.2 | 0.6 |
| Benz(a)anthrancene | 34 | 0.1 | 0.0 | 0.2 |
| Chrysene | 34 | 0.2 | 0.1 | 0.3 |
| Benzo(a)anthracene | 1 | 0.0 |  |  |
| Benzo(b)fluoranthene | 34 | 0.0 | 0.0 | 0.1 |



| | | | |
|---|---|---|---|
| Benzo(k)fluoranthene | 34 | 0.0 | 0.0 | 0.0 |
| Benzo(a)pyrene | 35 | 0.0 | 0.0 | 0.1 |
| Dibenzo(a,h)anthracene | 34 | 0.0 | 0.0 | 0.0 |
| Benzo(g,h,i)perylene | 34 | 0.0 | 0.0 | 0.0 |
| Indeno(1,2,3-c,d)pyrene | 34 | 0.0 | 0.0 | 0.0 |
| EPA 16 PAH | 1 | 6.5 | | |
| Total detected PAH | 35 | 8.3 | 6.0 | 10.5 |

**Table A3. Concentration of contaminants and nutrients in closed loop scrubber wash water. Note, that full references to sample analysis reports are given in the Supplementary material.**

| | #Samples included | Average concentration (µg/L) | 95% CI lower (µg/L) | 95% CI upper (µg/L) |
|---|---|---|---|---|
| Arsenic | 14 | 12.6 | 7.5 | 17.7 |
| Cadmium | 14 | 0.53 | 0.25 | 0.81 |
| Chromium | 7 | 2668 | -2179 | 7514 |
| Copper | 14 | 295 | 128 | 462 |
| Lead | 14 | 11.6 | 4.7 | 18.3 |
| Mercury | 9 | 0.07 | 0.05 | 0.10 |
| Nickel | 14 | 2794 | 1586 | 4001 |
| Vanadium | 14 | 11945 | 7254 | 16635 |
| Zinc | 14 | 479 | 127 | 831 |
| Naphthalene | 8 | 1.46 | 1.86 | -0.10 |
| Acenaphthylene | 7 | 0.03 | 0.02 | 0.01 |
| Acenaphthene | 7 | 0.21 | 0.16 | 0.06 |
| Fluorene | 7 | 0.74 | 0.66 | 0.13 |
| Phenanthrene | 8 | 3.16 | 2.82 | 0.80 |
| Anthracene | 7 | 0.04 | 0.04 | 0.00 |
| Flouranthene | 7 | 0.20 | 0.15 | 0.06 |
| Pyrene | 7 | 0.20 | 0.16 | 0.05 |
| Benz(a)anthrancene | 7 | 0.03 | 0.03 | 0.00 |
| Chrysene | 7 | 0.05 | 0.05 | 0.00 |
| Benzo(a)anthracene | 1 | 1.13 | | |
| Benzo(b)fluoranthene | 7 | 0.02 | 0.02 | 0.00 |
| Benzo(k)fluoranthene | 7 | 0.01 | 0.01 | 0.00 |
| Benzo(a)pyrene | 8 | 0.03 | 0.07 | -0.02 |
| Dibenzo(a,h)anthracene | 7 | 0.01 | 0.01 | 0.00 |
| Benzo(g,h,i)perylene | 7 | 0.01 | 0.01 | 0.00 |





| | | | | |
|---|---|---|---|---|
| Indeno(1,2,3-c,d)pyrene | 7 | 0.01 | 0.01 | 0.00 |
| Total detected PAH | 7 | 5.12 | 4.19 | 1.24 |





## Appendix B: Calculation examples for various discharges

**Calculation example for nutrients in sewage, grey water and food waste**

RoPax with 100 crew members and 1900 passenger capacity.
Average utilisation of passenger capacity: 50%
Daily production of Phophorus in sewage, per person: 1.6 grams
Daily production of Nitrogen in sewage, per person: 16 grams
Daily production of Phosphorus in grey water, per person: 1.9 grams
Daily production of Nitrogen in grey water, per person: 4.4 grams
Daily production of Phophorus in food waste, per person: 0.5 grams
Daily production of Nitrogen in food waste, per person: 1.7 grams
Duration of one trip: 2 h
Number of trips each day: 6
Daily travel time: 6 * 2 h = 12 h
Daily production of P: 100 crew * 24/24 h crew presence each day * (1.6 + 1.9 + 0.5) grams of P per each crew day + 1900 passengers * 0.5 passenger capacity utilisation * 12/24 passenger presence each day * (1.6 + 1.9 + 0.5) grams of P per each crew day  =  400 + 1900 g/day = 2300 grams/day
Annual production of P: 2300 g/day * 365 days = 840 kg/year ( and 4640 kg of N per year)

**Calculation example for antifouling paint residue release**

A calculation example for a vessel which operates in multiple sea areas is illustrated below. Only Cu(I)Oxide releases are included in this example:
Oil tanker, length 228m, breadth 32.2m, draught 14.3m coming from outside the Baltic Sea to Primorsk, Russia. Trip length 800 nautical miles, speed 15 knots, trip duration 53h. Wet surface area 9921 $m^2$(Wet surface calculated as described in Schneekluth & Bertram, "Ship Design for Efficiency and Economy, Butterworth-Heineman, 1998). Highest leaching rate of all the areas travelled is applied.
Leaching rate of Cu(I)pyrithione: 24.491 micrograms/($cm^2$ * day), 2.834 micrograms/($m^2$ * second)
Paint application factor, international value: 1
Biocide application factor, international values: 1 (Cu(I)Oxide )
During the trip, antifouling releases are:
Leaching rate * wet surface area * trip duration * paint application factor * biocide application factor
m(Cu(I)Oxide)= 9921 $m^2$ * 53 h * 3600 sec/h *2.834 microg/($m^2$ * sec) * 1 * 1 = 5.36 kg

**Calculation examples for a bilge water release from passenger and a cargo vessel**

Passenger ships:
Installed main engine power: 32580 kW
Bilge water production, l/day: 0.131284 l/kW *32580 kW + 373.416 l = 4651 litres/day
Oil: 4650 litres/day * 3228 micrograms/litre / 1000000 micrograms/gram= 15.012 grams/day
PAH16: 4650 litres/day * 58 micrograms/litre / 1000000 micrograms/gram= 0.270 grams/day
Surfactants: 4650 litres/day * 9 micrograms/litre / 1000000 micrograms/gram= 0.042 grams/day
Metals: 4650 litres/day * 545 micrograms/litre / 1000000 micrograms/gram= 2.535 grams/day

Cargo ships:
Installed main engine power: 10519 kW
Bilge water production, l/day: 0.024696 l/kW * 10519 kW + 154.4874 l = 414 litres/day
Oil: 414 litres/day * 3228 micrograms/litre / 1000000 micrograms/gram= 1.337 grams/day
PAH16: 414 litres/day * 58 micrograms/litre / 1000000 micrograms/gram= 0.024 grams/day
Surfactants: 414 litres/day * 9 micrograms/litre / 1000000 micrograms/gram= 0.004 grams/day
Metals: 414 litres/day * 545 micrograms/litre / 1000000 micrograms/gram= 0.226 grams/d