# Peer review of "Modeling of discharges from Baltic Sea shipping"

_Ocean Science, 2020_

## Referee Comment (RC1) · Burkard Watermann (Referee) · 5 Jan 2021

Please see my comments in the ms

Please also note the supplement to this comment:
https://os.copernicus.org/preprints/os-2020-99/os-2020-99-RC1-supplement.pdf

---

## Referee Comment (RC2) · Daniel Heydebreck (Referee) · 24 Jan 2021

The authors provided a well-written manuscript on a tool called STEAM that is able to calculate different streams of pollutants and nutrients from ships into the Baltic Sea. Detailed explanations are given in the Materials and Methods section and data from other studies and own surveys are provided in the supplement. STEAM has been a valuable tool before this update and now produces data that are even more useful. These data are valuable for scientists in different fields. To make STEAM data actually re-usable by other scientists, a detailed description of the model and publication is necessary – which was done by the authors. In addition, the output data have been persistently published with a DOI. Proper metadata is assigned to the published data and the data files comply with the CF Conventions. I consider this important so that

other scientist actually can re-use these data.

I do not see any major issues or inconsistencies in the manuscript with respect to scientific quality or methodology. There are a few question with respect to the Good Scientific Practice:

- Is the model code freely available?

- Is the model code well documented in the sense that it is re-usable by other scientists?

- Are the emission data available for additional years? If yes, are these data published as well? If not, is it possible to publish the other years? They might be valuable for

My specific comments mainly deal with missing references and figures that are hard to understand by color-blind readers. These comments are provided as comments in the attached pdf copy of the manuscript.

Please also note the supplement to this comment:
https://os.copernicus.org/preprints/os-2020-99/os-2020-99-RC2-supplement.pdf
* * *
[Figure]

[Figure]

[Figure]

**Modeling of discharges from Baltic Sea shipping**

Jukka-Pekka Jalkanen[1], Lasse Johansson[1], Magda Wilewska-Bien[2], Lena Granhag[2], Erik Ytreberg[2], K. Martin Eriksson[2,♣], Daniel Yngsell[2,♁], Ida-Maja Hassellöv[2], Kerstin Magnusson[3], Urmas Raudsepp[4], Ilja Maljutenko[4], Linda Styhre[5], Hulda Winnes[5] and Jana Moldanova[5]

5

[1]Atmospheric Composition, Finnish Meteorological Institute, Erik Palmen's Square 1, FI-00560 Helsinki, Finland
[2]Mechanics and Maritime Sciences, Chalmers University of Technology, Campus Lindholmen 41296 Gothenburg, Sweden
[3]IVL Swedish Environmental Research Institute, Lovén Center of Marine Sciences, Kristineberg, SE-451 78 Fiskebäckskil, Sweden
10 [4]Department of Marine Systems, Tallinn Technical University, Akademia Tee 15A, 12618 Tallinn, Estonia
[5]IVL Swedish Environmental Research Institute, Aschebergsgatan 44, 411 33 Göteborg, Sweden
♣ Current address: Gothenburg Centre for Sustainable Development (GMV), Aschebergsgatan 44, SE-41296 Gothenburg, Sweden
♁ Current address: The County Administrative Board of Västernorrland, SE-871 86 Härnösand, Sweden

15

*Correspondence to*: Jukka-Pekka Jalkanen (jukka-pekka.jalkanen@fmi.fi)

**Abstract.** This paper describes the new developments of the Ship Traffic Emission Assessment Model (STEAM) which enable modeling of pollutant discharges to water from ships. These include nutrients from black/grey water discharges as well as from
20 food waste. Further, also the modeling of contaminants in ballast, black, grey and scrubber water, bilge discharges and stern tube oil leaks are described, as well as releases of contaminants from antifouling paints. Each of the discharges are regulated by different sections of IMO MARPOL convention and emission patterns of different pollution releases vary significantly. The discharge patterns and total amounts for year 2012 in the Baltic Sea area are reported and open loop SOx scrubbing effluent was found to be the second largest pollutant stream by volume. The scrubber discharges have increased significantly
25 in recent years and their environmental impacts need to be investigated in detail.

**1. Introduction**

Ship operations produce waste streams related to propulsion and engine operations, as well as crew and passenger activities (Fig 1). The waste streams related to propulsion and engine operations include bilge water from the machinery spaces, stern
30 tube oil from lubrication of the propeller shaft, scrubber wash water from Exhaust Gas Cleaning Systems (EGCS) for reduction of emissions of sulphur oxides into the atmosphere, ballast water from maintaining ship stability, biocides used in antifouling paints to prevent hull growth, cooling water and tank cleaning residuals. Waste streams related to humans on board include food waste, black water (sewage), and water from galleys and showers (grey water), as well as other solid waste. Operational emissions and discharges from ships are regulated through international conventions, primarily the IMO MARPOL with its

**Fig. 1.**

---

## Author Comment (AC1) · 15 Mar 2021

We thank the reviewer for the feedback. The author response point by point, as well as the Track Changes version of the manuscript are provided as a supplement.

Please also note the supplement to this comment:
https://os.copernicus.org/preprints/os-2020-99/os-2020-99-AC1-supplement.zip

---

## Author Comment (AC2) · 15 Mar 2021

We thank the reviewer for the comments to our manuscript. The point by point response and Track Changes version of the manuscript are provided as supplements.

Please also note the supplement to this comment:
https://os.copernicus.org/preprints/os-2020-99/os-2020-99-AC2-supplement.zip

---

## Author Response (AR1)

**Author response to OS-2020-99 (Jalkanen et al)**

We thank both reviewers for their comments on our manuscript "Modeling of discharges from Baltic Sea shipping" by Jalkanen et al (OS-2020-99). Below is a point by point response to the feedback. The original comments from the reviewers are referenced with line numbers and author response is given in green text. Text additions to the manuscript are indicated with *italic*. Note, that line numbers used refer to the original version of the submitted manuscript.

The Reviewer 2 provided the following general comments:

- Is the model code freely available?
    - The STEAM model code is property of Finnish Meteorological Institute and it is not released as open source. The underlying principles of propulsion power estimation, emission factor assignment and algorithms used have been published in our earlier works. Assumptions used are reported in such manner that anyone building a corresponding model can implement them in detail.
- Is the model code well documented in the sense that it is re-usable by other scientists?
    - The model has a user manual, but it does not include description of module interfaces, provide a detailed technical documentation or 24/7 helpdesk to answer user questions. The time, effort and resourcing required to maintain these have not been considered necessary.
- Are the emission data available for additional years? If yes, are these data published as well? If not, is it possible to publish the other years? They might be valuable for
    - The annual Baltic Sea reporting of ship emissions to air, discharges to the sea and underwater noise are reported in the annual HELCOM Maritime meeting. We are working to release these environmental data for Baltic Sea shipping data, which covers the period starting from 2006 until present day. These will be deposited to data archives, like Zenodo, once the corresponding manuscript is published in a scientific journal series.

**Detailed comments**

**Response to reviewer1**

Line 139: "leaching"?

- Changed leaking to leaching

Line 145: "molluscs inducing pseudo-hermaphroditism and reproduction failure. As crustaceans..."

- Modified the sentence as suggested by the reviewer.

Line 353: "biocidal eroding or selfpolishing ..."
- Added biocidal eroding to antifouling paint section as suggested by the reviewer

Line 359: "The estimation of vessels with biocidal antifouling paint and non-biocidal hard coatings is a little confusing. If the percentages derived from the literature cited is correct, the percentage of of vessels without biocidal af would be quite high, 50% and 80% respectively. THis is in contradiction to the sentences 353 ff. As it is quite important to estimate more exactly the relation between leaching and non-leching under water coatings, the statement should be reasonable and has high impact on the calculations of biocide input."

- As written in the manuscript, the selection of antifouling paint depends on where the ship operates. As written in the section starting at line 353, a vessel operating in ice conditions may damage the antifouling paint when a polishing biocidal coating is used. Therefore, we assume that only 20% of the vessels that only operates in Gulf of Bothnia (the Baltic subbasin with the highest risk of ice conditions) are using biocidal antifouling paints. This implies that 80% of the vessels operating in this specific region only, are using a biocide-free icebreaker coating. For ships operating in the Baltic Sea and Gulf of Bothnia only, the assumption is that 50% are using biocide-free icebreaker coating while the other 50% are using biocidal coating. However, as we don't know if a specific vessel is actually using a biocidal coating or a biocide free icebreaker coating we multiplied the region-specific application factors (in this case 0.2 for Gulf of Bothnia, and 0.5 for Baltic Proper region) with the region-specific leaching rate to derive a generic leaching rate for the region-specific vessels

  We have now reprised this at line 365 and removed the sentence "In the present work, leaching rates are applied according to Table 3 (application factors included as described above)." And replaced it with the following:

  "*In the present work, we do not know if a specific vessel is using a biocidal coating or a biocide free icebreaker coating. Therefore, we multiplied the regional-specific application factors with the regional-specific leaching rate to derive a generic leaching rate for the region-specific vessels. As a consequence of the low application factor for ships operating in the Bothnian Bay only (20% of the vessels are assumed to be coated with antifouling paints), the generic average leaching rate of copper is significantly lower in the this region (3.1 µg/cm2/d) as compared to the International region (24.5 µg/cm2/d), where 100 % of the vessels are assumed to be coated with antifouling paint (Table 3).*"

Line 676: "The input of copper may still be underestimated as the release of copper by cathodic protection of niche areas like the cooling systems of vessels are out of scope. It is hardly impossible to estimate the release of copper by dissolution of anodes, but it should be addressed as an unsolved but relevant input factor."

- Comment of galvanic corrosion protection as a source of Cu and Zn to marine environment. We added reference to Rousseau et al (2009) and modified the end of Section 3.5 to:

  "*The approach taken in this manuscript (and STEAM model) allows future work with primarily*

*speed, salinity and temperature dependent antifouling paint releases. Maps for other antifouling paint residues can be found in Supplementary material. It should be noted, that this work does not evaluate the use of galvanic corrosion protection, which is also a source of copper and zinc to the marine environment (Rousseau et al., 2009). "*
* * *
**Response to Reviewer 2**

Line 35: "Please provide full references, here"

- Added references to IMO BWMC and AFS Conventions

Line 40: "Please provide a few references here (although the data quality is low)"
- Added references to classification society, country submissions and intergovernmental organisation reports as requested

Line 52: "Please also reference a few other publications by other authors that describe similar inventories. If nothing similar is available for the Baltic Sea, consider providing similar inventories for other marine regions -- e.g. global or North Sea shipping emissions."

- Added two references to ship discharge modeling work done for other parts of the world (Ivce et al, 2020; Seebens et al, 2013)

Line 103: "(persistent) organic pollutants as well; although they are not as well studied as classical air pollutants"

- We added volatile organic compounds (VOC) to this list of air pollutants. Persistent organic pollutants are included in this definition, because they are either in gaseous (VOC) or in organic carbon fraction of particulate matter (PM), depending on the volatility of the substance.

Line130: "grammar: "disinfection of the water"; question: however, do the other methods not disinfect the water? Which other method is applied here?"

- We have clarified the sentence to illustrate the physical/mechanical/chemical options to disinfect the ballast water.

Line 131: "Please add a reference here."

- Added reference to ballast water management system by-products (Werschkun et al, 2012) , as requested

Line 150: "Please consider adding two or three references here."

- Added reference to MEPC201.(62), which summarizes the Annex V rules

Line153: "Please describe the problem(s) caused by untreated sewage or black water in one or two sentences und provide a reference."

- We added the following:

*"Inefficient wastewater treatment may lead to bacterial and viral contamination of fisheries and public health risk.(Copeland, 2008)"*

Line 164: "... [description] of emissions [of environmental stressors] ..."

- We have modified this sentence to: "The aims of this study are to a) expand the existing STEAM ship emission model to include a description of environmental stressors from of shipping discharges to the marine environment."

Line 167. "please use different line style and line width for individual boundaries => simplify understanding for color-blind readers"

- We have changed the line thickness and color, to make the economic zones stand out better from other area definitions.

Line 174: "Please provide references if at least some extist."

- Added reference to Seebens et al (2013) and Ivce et al (2020)

Line 185: "second closing ")" is missing."

- Added closing parenthesis

Line 191: "Please point to the respective document in the supplement."

- Added reference to Supplementary material

Line 212: "Thanks for the honest statement and the possibility for other scientists to use the water volumes as basis for their work."

- We are happy to oblige. During this work it quickly became evident that toxicological analyses would require a very complicated chemical mix, and we felt that the chosen approach with discharge volumes and laboratory water analysis would be one way of meeting this requirement.

Line 229: "subscripts not properly rendered"

- Subscripts were corrected below Eqs 1a and 1b

Line 238: "Opening "(" missing"

- Redundant closing parenthesis was removed

Line 247: "… an [US EPA report] …"

- Added the article "a" to the sentence

Line 251: "Isn't it possible to get numbers on the sales of this type of oil and estimate the losses based on the sales?"

- Unfortunately, we could not find any statistics on the volume of stern tube oil lubricants sold globally. The only clue to stern tube oil release is to classification society documents which list daily lubricant consumption of six liters as normal.

Line 269: "formatting of reference"

- Fixed the references, which were partly redundant (from older document version)

Line 440: "In this context (or maybe also at another location) it might be reasonable to mention that plastic litter enters the oceans from ships -- although this is strictly controlled by MARPOL Annex VI. Since currently, marine litter is a hot research topic I would mention it. Mention it in the sense that plastic litter is another type of waste streams from ships to the ocean but that it is not considered here."

- We added one sentence of plastic litter to the end of the Introduction Section 1.7:
  *"Release of garbage, which contributes to plastic pollution of the seas, is prohibited, but not included in the current work, yet."*

Line 492: "Please consider using one specific color for each ship type amongst all figures."

- The vessel type portions of Figures 3, 4, 8, 9, 10 and 11 were revised to use uniform color scheme for different ship types.

Line 549, Line 554: "Please print the years 2012 and 2018 also into the two plots. Same as Fg. 5: please print the years into the two plots."

- We have added labels 2012 and 2018 to Figures 5 and 6

Line 596: "Color-blind people might not recognize the dots as they are easily mixed up with the background. The previous figures showing lines are OK: even if the actual colors cannot be distinguished, the emissions are easy to recognize because they are lines. In this figure, we have dots -- which are not as easy to differentiate from the background. Please consider removing the textured land background."

- The map background of Figure 8 was removed, and sea areas were painted in grey color to improve the contrast between the sea, land and points of ballast water release.

Line 614: "Does a ship load ballast water if it is full of cargo? If not, then many ship journeys would not transport ballast water -- maybe 50% because ships might travel with full cargo in one direction amd without cargo in the other. What is th authors' opinion on this?"

- Ballast water is used to balance/trim the vessel and to ensure that propeller is immersed.

Depending on the cargo weight and weight distribution, some of the ballast tanks may be filled as needed. How much of the ballast capacity is actually discharged depends on the cargo carried. It is difficult to determine how much and which tanks are being used. In the model, there is no connection to the data describing the weight of the cargo carried. This could be available from customs office data, though, but is not included in the current version of STEAM. Ships may also be filled with cargo (by volume), but with some light cargoes, the cargo capacity (in weight units) is not fully utilized. In these cases, ballast water is used to adjust the vessel draught.

Line 722: "Please provide explanation for abbreviations in the caption."

- Added explanations of abbreviations used to figure 12 caption

Line 754: "Maybe a reference for this? Personal communication would also be fine."

- Added reference to Kattner et al (2015) and EU report on compliance to Sulphur directive (2018)

Line 756: "maybe rather "after the start of the global 0.5% sulphur regulation." or "after the global 0.5% sulphur regulation will come into force."

- Modified the sentence as requested

Line 770: "Will the code also be freely available? If yes, where will it be availabe?"

- The STEAM model is the property of FMI and it is not released as open source code. Model outputs are shared with a wide audience in such a manner that individual ships cannot be identified. An accurate description of all the assumptions and algorithms is provided to ensure transparency of the modeling approach and the working principles.

Line 774: "well standardized files; next time, please consider uploading deflated netCDF4 files instead of zipped un-compressed netCDF files."

- Netcdf3 was selected to facilitate the use of STEAM output data in as many applications as possible. Some of the groups we have delivered STEAM data have had problems reading netcdf4 output. Netcdf4 output could be considered as a future development task, though.
* * *
Fixed typo at L450: Released -> release